# DISCOVERING MIXTURES OF STRUCTURAL CAUSAL MODELS FROM TIME SERIES DATA

## ABSTRACT

In fields such as finance, climate science, and neuroscience, inferring causal relationships from time series data poses a formidable challenge. While contemporary techniques can handle non-linear relationships between variables and flexible noise distributions, they rely on the simplifying assumption that data originates from the *same* underlying causal model. In this work, we relax this assumption and perform causal discovery from time series data originating from mixtures of *different* causal models. We infer both the underlying structural causal models and the posterior probability for each sample belonging to a specific mixture component. Our approach employs an end-to-end training process that maximizes an evidence-lower bound for data likelihood. Through extensive experimentation on both synthetic and real-world datasets, we demonstrate that our method surpasses state-of-the-art benchmarks in causal discovery tasks, particularly when the data emanates from diverse underlying causal graphs. Theoretically, we prove the identifiability of such a model under some mild assumptions.

## 1 INTRODUCTION

Causal discovery is the problem of discovering the underlying causal structure among observed variables in the data (Spirtes et al., 2000). It is a powerful tool to improve our understanding of the world. For instance, causal discovery algorithms can help uncover the relationships between various complex climatic phenomena from sea temperature measurements (Runge et al., 2019a). Time series data presents significant challenges to causal discovery: (1) Time series data often exhibits complex non-linear dependencies among both time steps and variables. (2) The space of all possible directed acyclic graphs (DAGs) increases super-exponentially with the number of time steps and variables (OEIS Foundation Inc., 2022). As a result, traditional causal discovery algorithms can be computationally demanding, limiting their scalability to large datasets. (3) Distinguishing between spurious correlations and true causal relationships is more difficult, especially in the context of high-dimensional time series data.

Several contemporary works address the problem of temporal causal discovery. One area of study is Granger causality (Granger, 1969), which is concerned with the forecastability of one time series given the other. (Tank et al., 2021) and (Khanna & Tan, 2019) use neural networks to infer Granger causality. However, Granger causality is not true causality (Peters et al., 2017); methods based on Granger causality cannot handle instantaneous effects and history-dependent noise. Another approach involves conditional independence testing (Runge et al., 2019b; Runge, 2020). It infers the causal skeleton by testing conditional independence between time series. Unfortunately, independence tests are computationally expensive and can only recover the causal graph up to a Markov equivalence class. Structural causal models (SCMs) explicitly model the relationship between different nodes. Approaches such as those proposed by (Hyvärinen et al., 2010) and (Pamfil et al., 2020) assume independent noise and linear relationships between variables to deduce the underlying causal graph. A more recent study by (Gong et al., 2022) employs neural networks to capture non-linear dependencies among variables and history-dependent noise.

However, all of these methods suffer from a crucial drawback – they assume the existence of a single underlying causal model that applies to the entire probability distribution. In reality, multi-modality is ubiquitous in the real world. The causal effects in a heterogeneous dataset cannot be captured accurately by a single SCM. We may need several distinct causal models on the same set of variables

to explain the observed data, even if different samples from the same distribution share similar causal mechanisms. For example, gene regulatory networks are particular to different cells at different developmental stages. But during experiments for cell lineage, one can only track the RNA expression levels of different cells with related but distinct gene regulatory networks, since every measurement destroys the cell (Qiu et al., 2022). Similarly, the stock market prices could have different causal graphs on different days, but we would still anticipate that their mutual influences follow similar patterns. Using a single causal model to explain the data can result in oversimplification and an inability to capture diverse causal mechanisms. Thus, it is imperative to account for this heterogeneity through multiple causal models to accurately represent the data distribution.

The task of discovering mixtures of causal graphs from observational time series data has received limited attention in the literature. Recent work, such as (Thiesson et al., 2013; Markham et al., 2022; Saeed et al., 2020; Zhou et al., 2022), have tackled the challenge of inferring causal models from mixture distributions. However, these approaches primarily focus on independent data and do not specifically address time series data. (Löwe et al., 2022) touched upon this problem by inferring a per-sample summary graph in an amortized framework, but their approach is limited to inferring Granger causal relationships and does not account for instantaneous effects.

This paper investigates a more realistic setting in which data is generated from different causal models. We assume that each time series comes from one out of $K$ possible, unknown underlying causal models. The membership of which time series comes from which causal model is also unknown. Our goal is to perform causal discovery by learning the complete SCMs as well as the corresponding membership for each time series sample. A complete SCM includes both the causal graph and its associated functional equations.

We tackle the problem of learning multiple SCMs from hetereogenous time series with our method, MCD. By optimizing an evidence lower bound for the data likelihood, we can infer both the complete SCM and the membership of each sample to its corresponding mixture component. Assuming the existence of 'representative' points whose membership to the clusters is known with a high degree of certainty, we characterize a sufficient condition for the identifiability of such mixture models.

In summary, our contributions are as follows:

- We tackle the realistic and challenging setting of discovering mixtures of structural causal models for time series data. We derive and optimize an evidence lower bound (ELBO) to simultaneously infer the underlying causal graphs, their associated functional equations, and the correspondence between each sample and the causal models.

- We show that under some mild assumptions about the existence of representative points, mixture distributions of identifiable causal models are identifiable. We also show the soundness of our ELBO objective by deriving its relationship with the true data likelihood.

- We demonstrate the empirical efficacy of our method, MCD, on both synthetic and real-world datasets. Notably, MCD can accurately assign samples to their underlying SCM and identify the corresponding causal graphs, even where the number of SCMs is mis-specified.

## 2 RELATED WORK

In this section, we classify causal discovery techniques based on the type of data they handle, distinguishing between independent data and time series data.

**Causal Discovery for independent Data.** Traditional causal discovery approaches can be roughly categorized as constraint-based or score-based. Constraint-based methods like PC (Spirtes et al., 2000) and FCI (Spirtes, 2001) algorithms infer the underlying causal graph up to the correct Markov Equivalence Class (MEC) by identifying conditional independence relations between the various observed variables. However, these methods rely on accurate conditional-independence testing which might not always be plausible. Furthermore, these methods suffer from identifiability issues within large equivalence classes (He et al., 2015). Score-based, such as the Fast Greedy Equivalence Search (FGES) (Chickering, 2002) and RL-BIC (Zhu et al., 2019), assign scores to potential directed acyclic graphs (DAGs) based on their ability to explain observational data. However, the large search space makes these algorithms inefficient in practice.

Recently, deep learning has been used as an effective tool for causal discovery and inference. (Ke et al., 2022) and (Lorch et al., 2022) learn to induce causal structures through supervised learning on synthetic datasets whose ground-truth graph is known. (Goudet et al., 2018) use a hill-climbing algorithm to orient the edges of a causal skeleton (obtained from other methods). They also learn the structural functional equations assuming that all the exogenous variables come from the same known distribution. Building upon (Pawlowski et al., 2020), (Geffner et al., 2022) extend their approach to learn both the underlying causal graph and the structural functional equations by using normalizing flow models to learn the distribution of the exogenous variables. They use the NOTEARS objective (Zheng et al., 2018) to enforce the acyclicity of the learned causal graph.

**Causal Discovery for Time Series Data.** Most works on time series causal discovery use the notion of Granger causality (Granger, 1969). (Tank et al., 2021) use component-wise Multi-Layer Perceptrons (cMLP) along with sparsity constraints on weight matrices in order to infer non-linear Granger causal links. (Khanna & Tan, 2019) use component-wise Statistical Recurrent Units (SRU), which incorporate single and multi-scale summary statistics from multi-variate time series for pairwise Granger causal detection. Amortized Causal Discovery (ACD) (Löwe et al., 2022) aims to infer Granger causality from time series data using a variational auto-encoder framework in conjuction with Graph Neural Networks (GNN). However, Granger causality is not true causality; it merely indicates the presence of an influencing relationship. Further, Granger causality cannot account for instantaneous effects, latent confounders, or history-dependent noise (Peters et al., 2017).

In contrast to Granger Causality, the framework of SCMs can theoretically model instantaneous effects, latent confounders, and history-dependent noise. (Hyvärinen et al., 2010) incorporate vector autoregressive models to the LiNGAM (Shimizu, 2014) algorithm to propose the VARLiNGAM algorithm for time series data. DYNOTEARS, proposed in (Pamfil et al., 2020), uses the NOTEARS DAG constraint (Zheng et al., 2018) to learn a Dynamic Bayesian Network. However, both VAR-LiNGAM and DYNOTEARS only account for linear causal relationships and do not account for history-dependent noise. (Runge et al., 2019b) extend the PC algorithm to time series data with the PCMCI method. PCMCI$^+$ (Runge, 2020) can handle instantaneous edges. (Malinsky & Spirtes, 2018) incorporate features of both constraint-based and score-based methods on multivariate time series data. (Yao et al., 2021) identify the underlying latent factors that influence variables and infer the causal relationships between them. (Gong et al., 2022) learns the time-lagged adjacency matrix given observational data while modeling the exogenous history-dependent noise distribution. However, all of these methods assume a single causal graph for the whole data distribution.

**Learning Multiple Causal Graphs.** Several contemporary works focus on the problem of causal discovery from heterogeneous independent data. (Thiesson et al., 2013) use a heuristic search-and-score method to learn the component DAG models. (Zhou et al., 2022) handle data coming from heterogeneous observational data by modeling the causal effects as functions of exogenous covariates. They can identify causal graphs with both hidden confounders and cyclic relationships. However, both of these methods only model linear causal relationships and Gaussian noise. (Saeed et al., 2020) use the FCI algorithm on mixture data to recover a composite representation of the mixture DAGs and use it to detect variables with varying conditional distributions across the components. (Strobl, 2019) infers causal structure from mixtures of DAGs, even when dealing with cycles, non-stationarity, non-linearity, latent variables, and selection bias, employing a conditional independence testing framework. (Markham et al., 2022) devise a kernel that measures similarities between the underlying non-linear causal structures of different samples. This similarity metric can be used to cluster points and subsequently perform causal discovery within each cluster.

(Huang et al., 2020) exploit non-stationarity in time series data to determine causal relationships using conditional independence tests. However, their setting differs from ours because they model the heterogeneity of causal mechanisms over time, rather than the heterogeneity of causal models across samples. Consequently, they do not infer separate causal models for different components. On the other hand, our method handles multi-modal time-series distributions with multiple underlying causal models. It learns one SCM per inferred cluster and the membership of each sample to the appropriate cluster while accommodating non-linear causal dependencies and history-dependent noise.

## 3 METHODOLOGY

**Preliminaries.** A Structural Causal Model (Pearl, 2009) (SCM) is a mathematical formalization of a data generative model that explicitly encodes the causal relationships between variables. Formally, an SCM over $D$ variables consists of a 5-tuple $\langle \mathcal{X}, \varepsilon, \mathcal{F}, \mathcal{G}, P(u) \rangle$

1. a set of endogenous (observed) variables $\mathcal{X} = \left\{ X^1, X^2, \ldots, X^D \right\}$;
2. a set of exogenous (noise) variables $\varepsilon = \left\{ \epsilon^1, \epsilon^2, \ldots, \epsilon^m \right\}$ which influence the endogenous variables; in general, $m \geq D$ since there could be latent confounders; but we assume causal sufficiency, i.e., that $m = D$.
3. a Directed Acyclic Graph (DAG) $\mathcal{G}$ denoting the causal links amongst the members of $\mathcal{X}$;
4. a set of $D$ functions $\mathcal{F} = \left\{ f^1, f^2, \ldots, f^D \right\}$ determining $\mathcal{X}$ through the equations $X^i = f^i(\mathrm{Pa}_{\mathcal{G}}^i, \epsilon^i), \mathrm{Pa}^i \subset \mathcal{X}, \epsilon^i \subset \varepsilon$, where $\mathrm{Pa}_{\mathcal{G}}^i$ denotes the parents of node $i$ in graph $\mathcal{G}$;
5. $P(\epsilon)$, which describes a distribution over noise $\epsilon$.

We assume that each sample $X$ generated by the SCM comes from a sample space $\mathbb{X}$. When $X$ consists of independent data, $\mathbb{X} \in \mathbb{R}^D$. For time series data, $\mathbb{X} \in \mathbb{R}^{D \times T}$, where $T$ is the number of timesteps. We can extend the notion of structural causal models to time-series data given a temporal causal graph $\mathcal{G}$ by describing the causal relationships as:

$$X_t^i = f_t^i(\mathrm{Pa}_{\mathcal{G}}^i(< t), \mathrm{Pa}_{\mathcal{G}}^i(t), \epsilon_t^i) \tag{1}$$

where $X_t^i$ denotes the value of the $i^{\text{th}}$ variable of the time-series at time $t$, $\mathrm{Pa}_{\mathcal{G}}^i(< t)$ denote the parents of node $i$ from the previous time-steps (i.e. lagged parents) and $\mathrm{Pa}_{\mathcal{G}}^i(t)$ denote the parents of node $i$ at the current time-step (i.e. instantaneous parents). In this work, however, we work with the additive noise model due to its structural identifiability Gong et al. (2022):

$$X_t^i = f_t^i(\mathrm{Pa}_{\mathcal{G}}^i(< t), \mathrm{Pa}_{\mathcal{G}}^i(t)) + \epsilon_t^i \tag{2}$$

**Problem Setting.** We are given $N$ examples of multi-variate time series with $D$ variables, each of length $T$, denoted by $\left\{ X_{1:T}^{1:D,(n)} \right\}_{n=1}^N$. We assume that each sample is generated in accordance with one of the $K$ (unknown) structural causal models $\mathcal{M}_{1:K}$. The problem statement is as follows:

*Given the time series samples $\left\{ X_{1:T}^{1:D,(n)} \right\}_{n=1}^N$, infer the $K$ unknown SCMs $\mathcal{M}_{1:K}$ that describe interactions occurring in a time window of length $L$.*

Each SCM $\mathcal{M}_i$ consists of both a graph, which is represented as an adjacency matrix $\mathcal{G}_i$ of size $(L+1) \times D \times D$, and its associated functional relationships. Our goal is to infer both the adjacency matrices for all $K$ SCMs and their functional equations in an unsupervised fashion.

Assuming a temporal causal structure with a fixed time lag is quite common, shared with Rhino (Gong et al., 2022), VARLiNGaM (Hyvärinen et al., 2010) and PCMCI (Runge et al., 2019b) amongst others. In practice, the time-lag $L$ is input as a hyperparameter.

### 3.1 MIXTURE CAUSAL DISCOVERY (MCD)

In this section, we detail our approach to learning mixtures of structural causal models from observational time-series data. We assume that the true data generation process follows the probabilistic graphical model shown in Figure 1.

We represent $K$ different SCMs as random variables $\mathcal{M}_{1:K}$. For each data sample indexed by $n$, we assign a categorical variable $Z^{(n)} \in \{1, \ldots, K\}$ to represents the membership of the SCM from which the data is generated. Mixtures of SCMs has the following generative process:

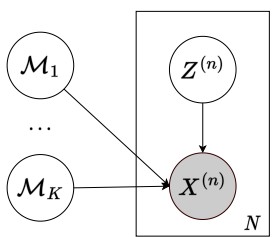

Figure 1: Probabilistic graphical model diagram of mixtures of SCMs. Shaded circles are observed variables and hollow circle are latent variables.

1. Choose $\mathcal{M}_{1:K} \sim p(\mathcal{M}_{1:K})$.
2. For each of the $N$ samples $X^{(n)}$:
   - Choose a mixture index $Z^{(n)} \sim p(Z)$.
   - Draw a time series $X^{(n)} \sim p(X \mid \mathcal{M}_{Z^{(n)}})$ from the the marginal distribution of the corresponding causal model.

Figure 2: Summary of model evaluation in the proposed method. Each sample has an associated variational distribution $r_\psi \left( Z^{(n)} | X^{(n)} \right)$ which designates its corresponding causal model out of the $K$ learned models $\mathcal{M}_{1:K}$. The likelihood $\log p_\theta \left( X^{(n)}; \mathcal{M}_{Z^{(n)}} \right)$ is evaluated with respect to the corresponding model $\mathcal{M}_{Z^{(n)}}$.

Thus, our goal is to infer the posterior distribution $p \left( \mathcal{M}_{1:K} \mid X \right)$ of the SCMs given the data samples. We model each SCM $\mathcal{M}_i$ as a pair $(\mathcal{G}_i, \theta_i)$, where $\mathcal{G}_i$ is the adjacency matrix, and $\theta_i$ represents the parameters of the neural network which approximates the functional relationships of the SCM.

We propose a variational inference framework to infer the parameters of the data generation process, since the true posterior $p \left( \mathcal{M}_{1:K} \mid X^{(1:N)} \right)$ is intractable. Thus, we derive and optimize an Evidence Lower Bound (ELBO) over parameters $(\theta, \phi, \psi)$ as:

$$\log p_\theta \left( X_{1:T}^{(1:N)} \right) \geq \sum_{n=1}^{N} \mathbb{E}_{q_\phi(\mathcal{M}_{1:K})} \left[ \mathbb{E}_{r_\psi \left( Z^{(n)} | X_{1:T}^{(n)} \right)} \left[ \log p_\theta \left( X_{1:T}^{(n)} \mid \mathcal{M}_{Z^{(n)}} \right) + \log p \left( Z^{(n)} \right) \right. \right.$$

$$\left. \left. + H \left( r_\psi \left( Z^{(n)} \mid X_{1:T}^{(n)} \right) \right) \right] \right] + \sum_{i=1}^{K} \mathbb{E}_{q_\phi(\mathcal{M}_i)} \left[ \log p(\mathcal{M}_i) + H \left( q_\phi(\mathcal{M}_i) \right) \right] \quad (3)$$

Here, $q_\phi \left( \mathcal{M}_i \right)$ represents the variational distribution of the causal model $\mathcal{M}_i$, and $r_\psi(Z^{(n)} \mid X^{(n)})$ represents the variational posterior distribution of the mixing rate for sample $X^{(n)}$. The number of causal models $K$ is a hyperparameter. $p(Z)$ represents our prior belief about the membership of samples to the causal models, typically considered to be a uniform distribution. For a detailed derivation, we refer the reader to Section A.1. Figure 2 shows a summary of how the likelihood is evaluated for every data sample, given the learned variational distributions and the mixing rates.

### 3.2 MODEL IMPLEMENTATION

In theory, any likelihood-based Bayesian causal structure learning algorithm, such as a time series-adapted variant of (Lorch et al., 2021), can be used to implement the loss terms in equation 3. We opted for the Rhino framework (Gong et al., 2022) with slight modifications to implement each of the $K$ causal models due to its ability to handle instantaneous effects and history-dependent noise. Hence, we share similar assumptions to Rhino; in particular, we require causal stationarity, causal minimality and causal sufficiency, in addition to some mild conditions on the likelihood function. For the sake of completeness, we mention these assumptions in Section A.2.

The main differences of MCD from Rhino arise from the need for modeling the variational distribution $r_\psi \left( Z^{(n)} \mid X^{(n)} \right)$ for the mixing rates. We parameterize it as a $K$-way categorical random variable and learn it separately for each sample. More precisely,

$$r_\psi \left( Z^{(n)} = k \mid X^{(n)} \right) = \frac{\exp \left( w_k^{(n)}/\tau_r \right)}{\sum_{k=1}^{K} \exp \left( w_k^{(n)}/\tau_r \right)}, \qquad k \in \{1, \ldots, K\},$$

where $w^{(n)} = \left[ w_1^{(n)}, \ldots, w_K^{(n)} \right] \in \mathbb{R}^K$ are learnable weight parameters for each sample and $\tau_r$ is a temperature hyperparameter. It is important to note that equation 3 requires an expectation over $r_\psi \left( Z^{(n)} \mid X^{(n)} \right)$, which we can evaluate exactly, unlike the need for a Monte-Carlo simulation over the variational distribution of the causal models $q_\phi \left( \mathcal{M}_{1:K} \right)$. Thus, we must compute the marginal likelihood of each sample under all $K$ causal models. This theoretically entails $K$ times more operations compared to Rhino during evaluation. In practice, we can calculate the marginal likelihoods over all causal models in a single forward pass by vectorizing the neural networks that

parameterize the causal models. Empirically, we observe that the computational complexity increase over Rhino leads to only a modest increase in run-time, much less than a factor of $K$. (Section B.5). We also implement weight sharing by utilizing embeddings $\theta_k$ to parameterize each causal model. These embeddings serve as inputs to hypernetworks that are shared across all $K$ causal models, implemented as neural networks. We refer the reader to Section C for a more detailed description of how MCD is implemented.

## 4 THEORETICAL ANALYSIS

In this section, we examine (1) a useful sufficient condition under which the described model is identifiable; (2) the relationship between the derived ELBO objective and the true data likelihood.

**Structural Identifiability.** We examine when the mixtures of SCM models are identifiable. We derive an intuitive sufficient condition for mixture model identifiability in terms of the existence of $K$ representative points from the sample space $\mathbb{X}$. These representative points exhibit a key characteristic: their association with a particular causal model is unequivocal, as determined by their scores computed from the marginal likelihood functions of the mixture components.

**Theorem 1** (Identifiability of finite mixture of causal models)**.** *Let $\mathcal{F}$ be a family of $K$ identifiable causal models, i.e. $\mathcal{F} = \left\{ \mathcal{L}_{\mathcal{M}}^{(i)} : \mathcal{M} \text{ is an identifiable causal model }, 1 \leq i \leq K \right\}$ and let $\mathcal{H}_K$ be the family of all $K-$finite mixtures of elements from $\mathcal{F}$, i.e.*

$$\mathcal{H}_K = \left\{ h : h = \sum_{i=1}^{K} \pi_i \mathcal{L}_{\mathcal{M}_i}, \mathcal{L}_{\mathcal{M}_i} \in \mathcal{F}, \pi_i > 0, \sum_{i=1}^{K} \pi_i = 1 \right\}$$

*where $\mathcal{L}_{\mathcal{M}_i}(x) = \sum_{\mathcal{M}} p(x \mid \mathcal{M}) p(\mathcal{M}_i = \mathcal{M})$ denotes the likelihood of $x$ evaluated with causal model $\mathcal{M}_i$. Further, assume that the following condition is met:*

$$\text{For every } i, 1 \leq i \leq K, \exists a_i \in \mathbb{X} \text{ such that } \frac{\mathcal{L}_{\mathcal{M}_i}(a_i)}{\sum_{j=1}^{K} \mathcal{L}_{\mathcal{M}_j}(a_i)} > \frac{1}{2}. \tag{*}$$

*Then the family $\mathcal{H}_K$ is identifiable, i.e., if $h_1 = \sum_{i=1}^{K} \pi_i \mathcal{L}_{\mathcal{M}_i}$ and $h_2 = \sum_{j=1}^{K} \pi'_j \mathcal{L}_{\mathcal{M}'_j} \in \mathcal{H}_K$ then:*

$$h_1 = h_2 \implies \forall i \in \{1, \ldots, K\} \ \exists j \in \{1, \ldots, K\} \text{ such that } \pi_i = \pi'_j \text{ and } \mathcal{M}_i = \mathcal{M}'_j.$$

(Relevant definitions and proof in Section A.3). To draw a parallel with clustering, this implies that for each cluster, there exists at least one point whose membership can be established with a high level of certainty to that specific cluster. Verifying this condition in practice is easy because it can be examined for individual sample points when an approximate likelihood function is learned for each mixture component, as is the case with our approach, MCD.

Furthermore, we note that as a direct consequence of the structural identifiability of the Rhino model Gong et al. (2022), a mixture of Rhino models is also structurally identifiable, provided that the assumptions in Section A.2 and condition $(*)$ are satisfied.

**Relationship between ELBO and Log Likelihood.** We verify the soundness of our derived ELBO objective in equation 3. By maximizing the ELBO, we can simultaneously learn the $K$ underlying causal graphs, their associated functional equations, and the membership of each sample to its respective causal model. We show that (Section A.4):

$$\log p_\theta(X) = \text{ELBO}(\theta, \phi, \psi) + \sum_{n=1}^{N} \mathbb{E}_{q_\phi(\mathcal{M}_{1:K})} \left[ \text{KL} \left( r_\psi \left( Z^{(n)} \mid X^{(n)} \right) \mid\mid p(Z^{(n)} \mid X^{(n)}, \mathcal{M}_{1:K}) \right) \right]$$
$$+ \text{KL} \left( q_\phi \left( \mathcal{M}_{1:K} \right) \mid\mid p \left( \mathcal{M}_{1:K} \mid X \right) \right).$$

Maximizing $\text{ELBO}(\theta, \phi, \psi)$ with respect to $(\theta, \phi, \psi)$ is equivalent to jointly (1) maximizing the log-likelihood $\log p_\theta(X)$ (2) minimizing the KL divergence between the true posterior $p \left( \mathcal{M}_{1:K} \mid X \right)$ and the variational distribution $q_\phi \left( \mathcal{M}_{1:K} \right)$; and (3) minimizing the expectation, under the variational distribution $q_\phi(\mathcal{M}_{1:K})$, of the KL divergence between the true posterior for model selection $p \left( Z^{(n)} \mid X^{(n)}, \mathcal{M}_{1:K} \right)$ and the variational posterior $r_\psi \left( Z^{(n)} \mid X^{(n)} \right)$ for each sample $X^{(n)}$.

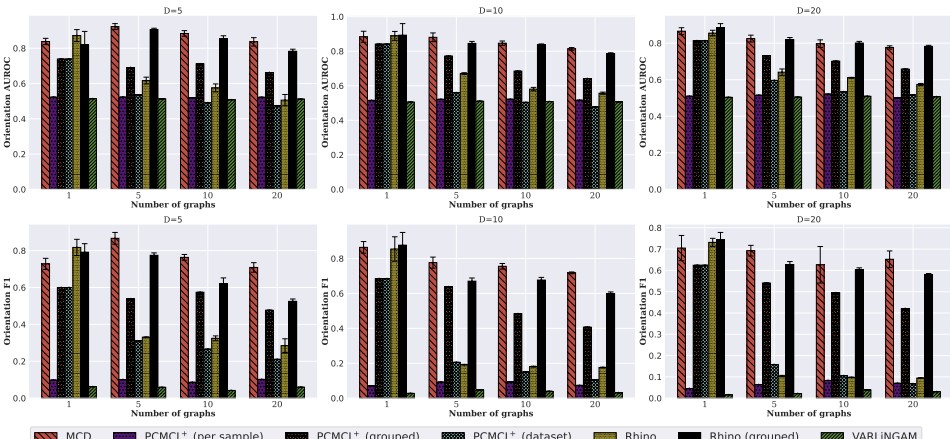

Figure 3: Results on the synthetic datasets for dimension $D = 5, 10, 20$. We present both the orientation AUROC (Area Under the Receiver Operating Characteristic) and F1 scores. (per sample) indicates that the baseline predicts one graph per sample, while (dataset) indicates that the baseline predicts one graph for the whole dataset. (grouped) signifies that the baseline was explicitly executed on samples clustered according to the true underlying causal graph. Average of 5 runs reported.

## 5 EXPERIMENTS

We experiment on both synthetic and real-world benchmark datasets. Our model was written in PyTorch and Lightning, and run on servers with Intel Xeon Gold 6230 CPUs and NVIDIA RTX 3090, RTX2080Ti, or A10 GPUs. In all our experiments, we train and validate the model on 100% of the data, since the variational posterior distribution $r_\psi \left( Z^{(n)} \mid X^{(n)} \right)$ is learned for each point. We pick the model with the lowest ELBO and evaluate the corresponding causal graphs.

### 5.1 EXPERIMENTAL SETUP

We benchmark against several state-of-the-art methods including Rhino (Gong et al., 2022), PCMCI$^+$ (Runge, 2020), DYNOTEARS (Pamfil et al., 2020) and VARLiNGaM (Hyvärinen et al., 2010). PCMCI$^+$ and DYNOTEARS can be used with two different options - one where the algorithm predicts one causal graph per sample and one where the algorithm predicts one graph to explain the whole dataset. We denote these options by suffixing the corresponding rows with (per sample) and (dataset) respectively. Further, we can also group examples by their true causal graph and predict one causal graph per group. This option is reported for PCMCI$^+$ and DYNOTEARS, and denoted by the suffix (grouped) in the results. Section D.2 details the steps for post-processing PCMCI$^+$'s output.

In practice, the number of mixture components is often unknown, which we treat as a hyperparameter. We use $K^*$ to denote the true number of SCMs, and $K$ to represent the input to MCD. In our experiments, we report the clustering accuracy for MCD in addition to traditional causal discovery metrics like orientation F1 score and AUROC (Area Under the Receiver Operator Curve). We refer the reader to Section C.1 for a detailed description of how the clustering accuracy is calculated.

### 5.2 DATASETS

**Synthetic datasets.** We generate a pool of $K^*$ random graphs (specifically, Erdős-Rényi graphs) and treat them as ground-truth causal graphs. To generate a sample $X^{(n)}$, we first randomly sample a graph $\mathcal{G}_k$ from this pool and use it to model relationships between variables using the equation:
$$X_t^{i,(n)} = f_k^i \left( \text{Pa}_{\mathcal{G}_k}^i(<t), \text{Pa}_{\mathcal{G}_k}^i(t) \right) + \epsilon_t^i.$$
The functional relationships $f_k^i$ between variables are represented by randomly initialized multi-layer perceptrons (MLPs), and the random noise $\epsilon_t^i$ is generated using history-conditioned quadratic spline flow functions (Durkan et al., 2019). We fix the number of variables $D$ and vary $K^*$ to be 1, 5, 10, and 20. We generate $N = 1000$ examples in the procedure described above. The time series length $T$ is 100, and the time lag $L$ is set to 2 for all the methods, which is also the value of lag used to simulate the data. For MCD, the number of mixture components $K$ is set to twice the number of true graphs (i.e., $K = 2K^*$) to showcase its robustness against over-specification of the

underlying number of components. We consider a uniform prior for the membership indicators $Z^{(n)}$, i.e. $p(Z = k) = \frac{1}{K} \; \forall k \in \{1, \dots, K\}$.

**Netsim Brain Connectivity.** The Netsim benchmark dataset (Smith et al., 2011) consists of simulated blood oxygenation level-dependent (BOLD) imaging data. Each variable represents a region of the brain, with the goal being to infer the interactions between the different regions. The dataset has 28 different simulations which differ in the number of variables and time-length over which the measurements are recorded. In our experiments, we consider two distinct setups:

In the first setup, we combine the time series which have length $T = 200$ and number of nodes $D = 5$ from simulations 1, 8, 10, 13, 14, 15, 16, 18, 21, 22, 23, and 24. This dataset comprises $N = 600$ samples, with $K^* = 14$ distinct underlying causal graphs. We refer to this setup as **Netsim**. This dataset exhibits significant graph membership imbalance, with the top 3 causal graphs accounting for 500 out of the 600 samples. Hence, we consider an exponentially weighted prior for the membership indicators, i.e. $p(Z = k) \propto \exp(-\lambda_p k) \; \forall k \in \{1, \dots, K\}$. We set $\lambda_p = 5$ and $K = 20$.

In the second setup, we consider the samples from simulation 3 comprising $N = 50$ time series, each with $D = 5$ nodes and $T = 200$ timepoints. These samples share the same ground-truth causal graph. We introduce heterogeneity by considering a pool of $K^* = 3$ random permutations and applying a randomly chosen one to the nodes of each sample and its corresponding ground truth causal graph. This setup is denoted as **Netsim-permuted**. We use a uniform prior for $p(Z)$ and set $K = 5$.

**DREAM3 Gene Network.** The DREAM3 dataset (Prill et al., 2010) is a real-world biology dataset consisting of measurements of gene expression levels obtained from yeast and E.coli cells. There are 5 distinct ground-truth networks, comprising 2 for E.coli, and 3 for Yeast, each with $D = 100$ nodes. Each time-series consists of $T = 21$ timesteps, with 46 trajectories recorded per graph. Thus, there are a total of $N = 230$ samples combined across all the networks. We mix samples from all 5 networks to simulate the scenario in which the identity of the cell from which the data is obtained is unknown. This is a challenging dataset due to the high dimensionality of the data and the small number of samples available for inferring causal relationships. We set the time lag $L = 2$ and $K = 10$.

Postprocessing model outputs for evaluating on Netsim and DREAM3 datasets is discussed in Section D.3.

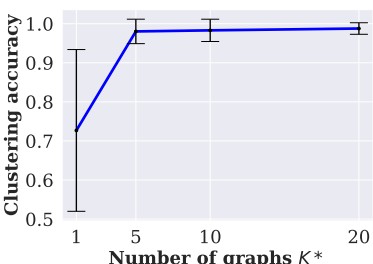

Figure 4: Clustering accuracy for MCD on the synthetic datasets (vs) the number of causal graphs $K^*$. The accuracy is averaged across 5 runs and across data dimensionality $D = 5, 10, 20$. Hyperparameter $K$ is set to $2K^*$ for all settings.

### 5.3 RESULTS

**Synthetic dataset.** We set the number of variables $D = 5, 10, 20$ and run our model on the synthetic dataset. Results are presented in Figure 3. We exclude results from DYNOTEARS due to its poor performance, where it mostly predicted zero matrices. We note that MCD handily outperforms the baseline methods in terms of orientation AUROC and F1 score when number of graphs $K^* = 5, 10, 20$. It achieves a comparable level of performance for $K^* = 1$ despite the misspecification of the number of models. Further, the gap between the F1 score for our method and the baselines widens as $K^*$ increases. Finally, our method outperforms PCMCI$^+$, Rhino and DYNOTEARS even when they are supplied with additional membership information $Z^{(n)}$ from which the corresponding samples $X^{(n)}$ are drawn.

We also report the clustering accuracy for MCD in Figure 4 for different values of $K^*$, averaged over the data dimensionalities $D = 5, 10, 20$. Remarkably, MCD achieves near perfect clustering for scenarios with multiple underlying graphs. The low clustering accuracy and relatively low F1 and AUROC scores for $K^* = 1$ are explained by the observation that MCD learns two similar mixture components to explain the single underlying mode in the distribution.

We also conduct further ablation studies on synthetic datasets and report results in Section B.1.

**Netsim Brain Connectivity.** The results on the Netsim dataset are presented in Table 1. In the first setup (**Netsim**), we observe that MCD is outperformed by the baselines PCMCI$^+$ and Rhino, even though they only predict one graph for the entire dataset. This is attributed to the similarity among

|  | Netsim | | Netsim-permuted | |
| --- | --- | --- | --- | --- |
| Method | Ori. AUROC($\uparrow$) | Ori. F1 ($\uparrow$) | Ori. AUROC($\uparrow$) | Ori. F1($\uparrow$) |
| PCMCI$^+$ (per sample) | 0.702 | 0.648 | 0.817 | **0.672** |
| PCMCI$^+$ (dataset) | 0.827 | **0.803** | 0.710 | 0.493 |
| PCMCI$^+$ (grouped) | 0.810 | 0.785 | 0.722 | 0.525 |
| VARLiNGAM | 0.638 | 0.598 | 0.781 | 0.604 |
| DYNOTEARS (per sample) | 0.706 | 0.588 | 0.850 | 0.276 |
| DYNOTEARS (dataset) | 0.674 | 0.626 | 0.826 | 0.451 |
| DYNOTEARS (grouped) | 0.629 | 0.584 | 0.848 | 0.459 |
| Rhino | **0.927 ± 0.008** | 0.585 ± 0.000 | 0.873 ± 0.007 | 0.530 ± 0.020 |
| MCD (ours) | 0.807 ± 0.006 | 0.680 ± 0.012 | **0.929 ± 0.018** | 0.641 ± 0.024 |

Table 1: Results on the Netsim and Netsim-permuted datasets. (per sample) indicates that the baseline predicts one graph per sample, while (dataset) indicates that the baseline predicts one graph for the whole dataset. (grouped) signifies that the baseline was explicitly executed on samples clustered according to the true underlying causal graph. MCD achieves a clustering accuracy of $61.40 \pm 1.66\%$ on Netsim and $87.84 \pm 17.7\%$ on Netsim-permuted.

| Method | Ori. AUROC ($\uparrow$) | Ori. F1 ($\uparrow$) |
| --- | --- | --- |
| PCMCI$^+$ (per sample) | 0.500 | 0.008 |
| PCMCI$^+$ (grouped) | 0.513 | 0.052 |
| DYNOTEARS (per sample) | 0.505 | 0.031 |
| DYNOTEARS (dataset) | 0.504 | 0.033 |
| DYNOTEARS (grouped) | 0.504 | 0.033 |
| Rhino | 0.527 ± 0.004 | 0.057 ± 0.003 |
| MCD (ours) | **0.555 ± 0.003** | **0.133 ± 0.009** |

Table 2: Results on the DREAM3 dataset. (per sample) indicates that the baseline predicts one graph per sample, while (dataset) indicates that the baseline predicts one graph for the whole dataset. (grouped) signifies that the baseline was explicitly executed on samples clustered according to the true underlying causal graph. MCD achieves a clustering accuracy of $94.83 \pm 4.49\%$.

the various underlying graphs in the Netsim dataset and the strong imbalance in the data. Our model faces sample complexity issues because it learns multiple causal graphs, whereas other methods perform reasonably well by predicting only one. This highlights the idea that learning a mixture model is only beneficial when the underlying SCMs differ from one another significantly. In such a scenario, the benefits of learning multiple graphs outweigh the drawbacks of limited samples per model. This explanation is also supported by the observation that PCMCI$^+$ (grouped) achieves lower performance than its single graph counterpart. Further, MCD achieves a relatively low clustering accuracy of $61.40 \pm 1.66\%$, due to the inherent similarities in the underlying SCMs.

In the second setup (**Netsim-permuted**), MCD outperforms all baselines in terms of AUROC, achieving a $6.4\%$ higher score than the next best baseline, and obtains the second highest F1 score. This setting illustrates the benefits of modeling heterogeneity, even when it comes from a simple permutation of nodes. In this setting, MCD achieves a clustering accuracy of $87.84 \pm 17.7\%$, highlighting its ability to accurately group samples when the underlying causal models are sufficiently diverse.

**DREAM3 Gene Network.** The results on the DREAM3 dataset are presented in Table 2. Expectedly, all methods face significant challenges in accurately inferring the causal relationships. However, out of all the considered baselines, MCD achieves the most promising performance both in terms of AUROC and F1 score. It is especially encouraging that MCD is able to accurately cluster samples by their causal models, with a remarkable clustering accuracy of $94.83 \pm 4.49\%$.

## 6 CONCLUSION AND DISCUSSION

In this work, we examined the problem of discovering mixtures of structural causal models from time series data. This is a problem with far-reaching applications in climate, finance, and healthcare, among other fields, since multimodal and hetereogeneous data is ubiquitous in practice. We proposed an end-to-end deep-learning method to infer both the underlying SCMs and the mixture component membership of each sample. We discussed the structural identifiability of our model and demonstrated the empirical efficacy of our method on both synthetic and real-world datasets generated from diverse component SCMs. Future work could tackle latent confounders and non-stationarity in time.

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

# A  THEORY

## A.1  ELBO DERIVATION

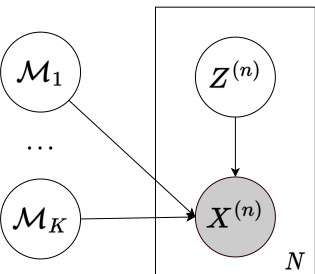

Figure 5: The assumed data generation model. First, the mixture index $Z^{(n)}$ is drawn from a $K$-way categorical distribution ($Z^{(n)} \sim \text{Cat}(K), Z^{(n)} \in \{1, \ldots, K\}$), and a causal model is drawn from the corresponding mixture component distribution $\mathcal{M} \sim p\left(\mathcal{M}_{Z^{(n)}}\right)$. A sample $X^{(n)}$ is then drawn in accordance with the chosen causal model $\mathcal{M}$.

Denote the causal models as $\mathcal{M}_{1:K} = (\mathcal{M}_1, \ldots, \mathcal{M}_K)$ and the sample $X = \left\{X^{(n)}\right\}_{n=1}^{N}$. Then, we can write the log-likelihood under the assumed model as follows:

$$
\begin{aligned}
\log p_\theta(X) &= \log \left[ \sum_{\mathcal{M}_{1:K}} p_\theta\left(X \mid \mathcal{M}_{1:K}\right) p(\mathcal{M}_{1:K}) \times \frac{q_\phi(\mathcal{M}_{1:K})}{q_\phi(\mathcal{M}_{1:K})} \right] \\
&= \log \mathbb{E}_{q_\phi(\mathcal{M}_{1:K})} \left[ \frac{p_\theta\left(X \mid \mathcal{M}_{1:K}\right) p(\mathcal{M}_{1:K})}{q_\phi(\mathcal{M}_{1:K})} \right] \\
&\geq \mathbb{E}_{q_\phi(\mathcal{M}_{1:K})} \left[ \log \frac{p_\theta\left(X \mid \mathcal{M}_{1:K}\right) p(\mathcal{M}_{1:K})}{q_\phi(\mathcal{M}_{1:K})} \right] \quad \text{(using Jensen's inequality)} \\
&= \mathbb{E}_{q_\phi(\mathcal{M}_{1:K})} \left[ \log p_\theta\left(X \mid \mathcal{M}_{1:K}\right) + \log p(\mathcal{M}_{1:K}) - \log q_\phi(\mathcal{M}_{1:K}) \right]
\end{aligned}
$$

Since the sample points are conditionally independent given the causal models, we can write:

$$
\begin{aligned}
\log p_\theta(X) \geq \sum_{n=1}^{N} \mathbb{E}_{q_\phi(\mathcal{M}_{1:K})} \left[ \log p_\theta\left(X^{(n)} \mid \mathcal{M}_{1:K}\right) \right] \\
+ \mathbb{E}_{q_\phi(\mathcal{M}_{1:K})} \left[ \log p(\mathcal{M}_{1:K}) - \log q_\phi(\mathcal{M}_{1:K}) \right]
\end{aligned}
$$

Further, note that:

$$\log p_\theta(X^{(n)} \mid \mathcal{M}_{1:K}) = \log \left[ \sum_{Z^{(n)}} p_\theta(X^{(n)} \mid Z^{(n)}, \mathcal{M}_{1:K}) p(Z^{(n)} \mid \mathcal{M}_{1:K}) \right]$$

$$= \log \left[ \sum_{Z^{(n)}} p_\theta(X^{(n)} \mid Z^{(n)}, \mathcal{M}_{1:K}) p(Z^{(n)}) \times \frac{r_\psi \left( Z^{(n)} \mid X^{(n)} \right)}{r_\psi \left( Z^{(n)} \mid X^{(n)} \right)} \right]$$

$$= \log \mathbb{E}_{r_\psi(Z^{(n)} \mid X^{(n)})} \left[ \frac{p_\theta(X^{(n)} \mid Z^{(n)}, \mathcal{M}_{1:K}) p(Z^{(n)})}{r_\psi \left( Z^{(n)} \mid X^{(n)} \right)} \right]$$

$$\geq \mathbb{E}_{r_\psi \left( Z^{(n)} \mid X^{(n)} \right)} \left[ \log \frac{p_\theta(X^{(n)} \mid Z^{(n)}, \mathcal{M}_{1:K}) p(Z^{(n)})}{r_\psi \left( Z^{(n)} \mid X^{(n)} \right)} \right] \quad \text{(using Jensen's inequality)}$$

$$= \mathbb{E}_{r_\psi \left( Z^{(n)} \mid X^{(n)} \right)} \left[ \log p_\theta(X^{(n)} \mid Z^{(n)}, \mathcal{M}_{1:K}) + \log p(Z^{(n)}) - \log r_\psi \left( Z^{(n)} \mid X^{(n)} \right) \right].$$

Further, we assume that $p_\theta(X^{(n)} \mid Z^{(n)}, \mathcal{M}_{1:K}) = p_\theta \left( X^{(n)} \mid \mathcal{M}_{Z^{(n)}} \right)$. Putting it all together, and using the independence of the causal models, we obtain:

$$\log p_\theta(X) \geq \sum_{n=1}^{N} \mathbb{E}_{q_\phi(\mathcal{M}_{1:K})} \left[ \mathbb{E}_{r_\psi \left( Z^{(n)} \mid X^{(n)} \right)} \left[ \log p_\theta(X^{(n)} \mid \mathcal{M}_{Z^{(n)}}) + \log p(Z^{(n)}) - \log r_\psi \left( Z^{(n)} \mid X^{(n)} \right) \right] \right]$$

$$+ \sum_{i=1}^{K} \mathbb{E}_{q_\phi(\mathcal{M}_i)} \left[ \log p(\mathcal{M}_i) - \log q_\phi(\mathcal{M}_i) \right]$$

$$\equiv \text{ELBO}(\theta, \phi, \psi)$$

## A.2 THEORETICAL ASSUMPTIONS

In this section, we list out the theoretical assumptions used in Rhino (Gong et al., 2022); our model also operates under similar assumptions for each mixture component, since we implement the component SCMs as Rhino models.

**Assumption 1** (Causal Stationarity). (Runge, 2018) The time series $X$ with a graph $G$ is called causally stationary over a time index set $\mathcal{T}$ if and only if for all links $X_{t-\tau}^i \to X_t^j$ in the graph

$$X_{t-\tau}^i \not\perp\!\!\!\perp X_t^j \mid X_t \backslash \left\{ X_{t-\tau}^i \right\} \qquad \text{holds for all } t \in \mathcal{T}.$$

Informally, this assumption states that the causal graph does not change over time, i.e., the resulting time series is stationary.

**Assumption 2** (Causal Markov Property). (Peters et al., 2017) Given a DAG $G$ and a probability distribution $p$, $p$ is said to satisfy the causal Markov property, if it factorizes according to $G$, i.e. $p(x) = \prod_{i=1}^{D} p \left( x_i \mid \text{Pa}_G^i(x_i) \right)$. In other words, each variable is independent of its non-descendent given its parents.

**Assumption 3** (Causal Minimality). Given a DAG $G$ and a probability distribution $p$, $p$ is said to satisfy the causal minimality with respect to $G$, if $p$ is Markovian with respect to $G$ but not to any proper subgraph of $G$.

**Assumption 4** (Causal Sufficiency). A set of observed variables $V$ is said to be causally sufficient for a process $X_t$ if, in the process, every common cause of two or more variables in $V$ is also in $V$, or is constant for all units in the population. In other words, causal sufficiency implies the absence of latent confounders in the data.

**Assumption 5** (Well-defined Density). The likelihood of each mixture component (i.e. the likelihood function of each Rhino model) is absolutely continuous with respect to a Lebesgue or counting measure and $|\log p \left( X_{0:T}; G \right)| < \infty$ for all possible $G$.

A.3 IDENTIFIABILITY OF THE MIXTURE OF CAUSAL MODELS

**Definition 1** (Identifiability). Let $P = \{p_\theta : \theta \in \mathcal{T}\}$ be a family of distributions, each member of which is parameterized by the parameter $\theta$ from a parameter space $\mathcal{T}$. Then $P$ is said to be identifiable if

$$p_{\theta_1} = p_{\theta_2} \implies \theta_1 = \theta_2 \quad \forall \theta_1, \theta_2 \in \mathcal{T}.$$

**Definition 2** (Identifiability of finite mixtures). Let $\mathcal{F}$ be a family of distributions. The family of $K-$mixture distributions on $\mathcal{F}$, defined as $\mathcal{H}_K = \left\{ h : h = \sum_{i=1}^K \pi_i f_i, f_i \in \mathcal{F}, \pi_i > 0, \sum_{i=1}^K \pi_i = 1 \right\}$, is said to be identifiable if

$$\sum_{i=1}^K \pi_i f_i = \sum_{j=1}^K \pi_j' f_j' \implies \forall i \, \exists j \text{ such that } \pi_i = \pi_j' \text{ and } f_i = f_j'.$$

Here, we quote a result from (Yakowitz & Spragins, 1968) that established a necessary and sufficient condition for the identifiability of finite mixtures of multivariate distributions.

**Theorem 2** ((Yakowitz & Spragins, 1968) Identifiability of finite mixtures of distributions). *Let $\mathcal{F} = \{F(x; \alpha), \alpha \in \mathbb{R}^m, x \in \mathbb{R}^n\}$ be a finite mixture of distributions. Then $\mathcal{F}$ is identifiable if and only if $\mathcal{F}$ is a linearly independent set over the field of real numbers.*

In other words, this theorem states that a mixture of distributions is identifiable if and only if none of the individual mixture components can be expressed as a mixture of distributions from the same family. However, it can be difficult to reason about such a condition in practice, since the precise form of the marginal likelihood functions is rarely known. Conversely, the likelihood can be evaluated quite easily on discrete points, at least approximately if not exactly.

Here, we describe a sufficient condition for the identifiability of finite mixtures of *identifiable* causal models.

**Theorem 1** (Identifiability of finite mixture of causal models). *Let $\mathcal{F}$ be a family of $K$ identifiable causal models, i.e. $\mathcal{F} = \left\{ \mathcal{L}_\mathcal{M}^{(i)} : \mathcal{M} \text{ is an identifiable causal model}, 1 \leq i \leq K \right\}$ and let $\mathcal{H}_K$ be the family of all $K-$finite mixtures of elements from $\mathcal{F}$, i.e.*

$$\mathcal{H}_K = \left\{ h : h = \sum_{i=1}^K \pi_i \mathcal{L}_{\mathcal{M}_i}, \mathcal{L}_{\mathcal{M}_i} \in \mathcal{F}, \pi_i > 0, \sum_{i=1}^K \pi_i = 1 \right\}$$

*where $\mathcal{L}_{\mathcal{M}_i}(x) = \sum_\mathcal{M} p(x \mid \mathcal{M}) p(\mathcal{M}_i = \mathcal{M})$ denotes the likelihood of $x$ evaluated with causal model $\mathcal{M}_i$. Further, assume that the following condition is met:*

$$\textit{For every } i, 1 \leq i \leq K, \exists a_i \in \mathbb{X} \textit{ such that } \frac{\mathcal{L}_{\mathcal{M}_i}(a_i)}{\sum_{j=1}^K \mathcal{L}_{\mathcal{M}_j}(a_i)} > \frac{1}{2}. \qquad (*)$$

*Then the family $\mathcal{H}_K$ is identifiable, i.e., if $h_1 = \sum_{i=1}^K \pi_i \mathcal{L}_{\mathcal{M}_i}$ and $h_2 = \sum_{j=1}^K \pi_j' \mathcal{L}_{\mathcal{M}_j'} \in \mathcal{H}_K$ then:*

$$h_1 = h_2 \implies \forall i \in \{1, \ldots, K\} \, \exists j \in \{1, \ldots, K\} \textit{ such that } \pi_i = \pi_j' \textit{ and } \mathcal{M}_i = \mathcal{M}_j'.$$

*Proof.* From Theorem 2, we have that $\mathcal{H}_K$ is identifiable if and only if for any $\alpha_1, \ldots, \alpha_K \in \mathbb{R}$,

$$\sum_{j=1}^K \alpha_j \mathcal{L}_{\mathcal{M}_j} = 0 \implies \alpha_j = 0 \quad \forall j \in \{1, \ldots, K\}$$

Note that $\sum_{j=1}^K \alpha_j \mathcal{L}_{\mathcal{M}_j} = 0 \implies \sum_{j=1}^K \alpha_j \mathcal{L}_{\mathcal{M}_j}(x) = 0 \quad \forall x \in \mathbb{X}$. In particular,

$$\sum_{j=1}^K \alpha_j \mathcal{L}_{\mathcal{M}_j}(a_i) = 0 \quad \forall i \in \{1, \ldots, K\}, \qquad (4)$$

where $a_i$ is as defined in Condition $(*)$. Denote $\mathcal{L}_{\mathcal{M}_j}(a_i) = \beta_{ij}$. Then Equation 4 can be written as:

$$\begin{bmatrix} \beta_{11} & \dots & \beta_{1K} \\ \vdots & & \vdots \\ \beta_{K1} & \dots & \beta_{KK} \end{bmatrix} \begin{bmatrix} \alpha_1 \\ \vdots \\ \alpha_K \end{bmatrix} = \mathbf{0}. \tag{5}$$

Or equivalently

$$\boldsymbol{\beta}\boldsymbol{\alpha} = \mathbf{0}. \tag{6}$$

Note that $\boldsymbol{\alpha} = 0$ if and only if $\boldsymbol{\beta}$ is full rank. We now show that Condition $(*)$ implies that $\boldsymbol{\beta}$ is strictly diagonally dominant and hence full rank. Note that Condition $(*)$ can be equivalently written as:

$$\frac{\beta_{ii}}{\sum_{j=1}^{K} \beta_{ij}} > \frac{1}{2} \implies 2\beta_{ii} > \sum_{j=1}^{K} \beta_{ij}$$

$$\implies \beta_{ii} > \sum_{j=1, j\neq i}^{K} \beta_{ij}$$

which implies strict diagonal dominance since $\beta_{ij} > 0$. Hence $\boldsymbol{\alpha} = 0$ thus implying linear independence. $\qquad\square$

Note that $a_i$ refers to any point in the support of the mixture distribution such that the condition * is satisfied. It is not a 'sample' from the $i^{\text{th}}$ SCM in the sense that it is not randomly sampled from the SCM, but it can be potentially explicitly chosen for the condition to hold.

## A.4 RELATIONSHIP BETWEEN ELBO AND LOG-LIKELIHOOD

In this section, we derive an exact relationship between the derived evidence lower bound $\text{ELBO}(\theta, \phi, \psi)$ and the log-likelihood $\log p_\theta(X)$.

First, note that:

$$p_\theta(X)p\left(\mathcal{M}_{1:K} \mid x\right) = p_\theta\left(X \mid \mathcal{M}_{1:K}\right)p\left(\mathcal{M}_{1:K}\right)$$

and hence:

$$p_\theta(X) = \frac{p_\theta\left(X \mid \mathcal{M}_{1:K}\right)p\left(\mathcal{M}_{1:K}\right)}{p\left(\mathcal{M}_{1:K} \mid X\right)}.$$

The log-likelihood can be written as:

$$\begin{aligned}
\log p_\theta(X) &= \mathbb{E}_{q_\phi(\mathcal{M}_{1:K})}\left[\log p_\theta(X)\right] \\
&= \mathbb{E}_{q_\phi(\mathcal{M}_{1:K})}\left[\log \frac{p_\theta\left(X \mid \mathcal{M}_{1:K}\right)p\left(\mathcal{M}_{1:K}\right)}{p\left(\mathcal{M}_{1:K} \mid X\right)} \times \frac{q_\phi(\mathcal{M}_{1:K})}{q_\phi(\mathcal{M}_{1:K})}\right] \\
&= \mathbb{E}_{q_\phi(\mathcal{M}_{1:K})}\left[\log p_\theta\left(X \mid \mathcal{M}_{1:K}\right) + \log p\left(\mathcal{M}_{1:K}\right)\right] \\
&\quad + \sum_{i=1}^{K}\mathbf{H}\left(q_\phi(\mathcal{M}_i)\right) + \text{KL}\left(q_\phi\left(\mathcal{M}_{1:K}\right) \,\|\, p\left(\mathcal{M}_{1:K} \mid X\right)\right) \\
&= \mathbb{E}_{q_\phi(\mathcal{M}_{1:K})}\left[\sum_{n=1}^{N}\log p_\theta\left(X^{(n)} \mid \mathcal{M}_{1:K}\right) + \sum_{i=1}^{K}\log p\left(\mathcal{M}_i\right)\right] \\
&\quad + \sum_{i=1}^{K}\mathbf{H}\left(q_\phi(\mathcal{M}_i)\right) + \text{KL}\left(q_\phi\left(\mathcal{M}_{1:K}\right) \,\|\, p\left(\mathcal{M}_{1:K} \mid X\right)\right)
\end{aligned}$$

Also note that, using the rules of conditional probability:

$$\frac{p_\theta(X^{(n)} \mid \mathcal{M}_{1:K})}{p_\theta(X^{(n)} \mid \mathcal{M}_{1:K}, Z^{(n)})} = \frac{p_\theta(X^{(n)}, \mathcal{M}_{1:K})}{p(\mathcal{M}_{1:K})} \times \frac{p(Z^{(n)}, \mathcal{M}_{1:K})}{p_\theta(X^{(n)}, Z^{(n)}, \mathcal{M}_{1:K})}$$

$$= \frac{p(Z^{(n)} \mid \mathcal{M}_{1:K})}{p(Z^{(n)} \mid X^{(n)}, \mathcal{M}_{1:K})}$$

$$= \frac{p(Z^{(n)})}{p(Z^{(n)} \mid X^{(n)}, \mathcal{M}_{1:K})}$$

where the last step follows from the fact that $Z^{(n)}$ and $\mathcal{M}_i$ are independent.

Thus, we can write:

$$p_\theta(X^{(n)} \mid \mathcal{M}_{1:K}) = \mathbb{E}_{r_\psi(Z^{(n)} \mid X^{(n)})} \left[ p_\theta(X^{(n)} \mid \mathcal{M}_{1:K}) \right]$$

$$= \mathbb{E}_{r_\psi(Z^{(n)} \mid X^{(n)})} \left[ \frac{p_\theta(X^{(n)} \mid \mathcal{M}_{1:K}, Z^{(n)}) p(Z^{(n)})}{p(Z^{(n)} \mid X^{(n)}, \mathcal{M}_{1:K})} \right]$$

$$= \mathbb{E}_{r_\psi(Z^{(n)} \mid X^{(n)})} \left[ \frac{p_\theta(X^{(n)} \mid \mathcal{M}_{Z^{(n)}}) p(Z^{(n)})}{p(Z^{(n)} \mid X^{(n)}, \mathcal{M}_{1:K})} \times \frac{r_\psi\left(Z^{(n)} \mid X^{(n)}\right)}{r_\psi\left(Z^{(n)} \mid X^{(n)}\right)} \right].$$

Thus,

$$\log p_\theta(X) = \mathbb{E}_{q_\phi(\mathcal{M}_{1:K})} \left[ \sum_{n=1}^{N} \mathbb{E}_{r_\psi(Z^{(n)} \mid X^{(n)})} \left[ \log p_\theta\left(X^{(n)} \mid \mathcal{M}_{Z^{(n)}}\right) + \log p(Z^{(n)}) \right] + \mathrm{H}\left(r_\psi\left(Z^{(n)} \mid X^{(n)}\right)\right) \right.$$

$$\left. + \mathrm{KL}\left(r_\psi(Z^{(n)} \mid X^{(n)}) \,\|\, p(Z^{(n)} \mid X^{(n)}, \mathcal{M}_{1:K})\right) + \sum_{i=1}^{K} \log p\left(\mathcal{M}_i\right) \right] + \sum_{i=1}^{K} \mathrm{H}\left(q_\phi(\mathcal{M}_i)\right)$$

$$+ \mathrm{KL}\left(q_\phi\left(\mathcal{M}_{1:K}\right) \,\|\, p\left(\mathcal{M}_{1:K} \mid X\right)\right).$$

Noting that

$$\mathrm{ELBO}(\theta, \phi, \psi) \equiv \sum_{n=1}^{N} \mathbb{E}_{q_\phi(\mathcal{M}_{1:K})} \left[ \mathbb{E}_{r_\psi(Z^{(n)} \mid X^{(n)})} \left[ \log p_\theta(X^{(n)} \mid \mathcal{M}_{Z^{(n)}}) + \log p(Z^{(n)}) - \log r_\psi\left(Z^{(n)} \mid X^{(n)}\right) \right] \right]$$

$$+ \sum_{i=1}^{K} \mathbb{E}_{q_\phi(\mathcal{M}_i)} \left[ \log p(\mathcal{M}_i) - \log q_\phi(\mathcal{M}_i) \right]$$

we obtain that:

$$\log p_\theta(X) = \mathrm{ELBO}(\theta, \phi, \psi) + \sum_{n=1}^{N} \mathbb{E}_{q_\phi(\mathcal{M}_{1:K})} \left[ \mathrm{KL}\left(r_\psi\left(Z^{(n)} \mid X^{(n)}\right) \,\|\, p(Z^{(n)} \mid X^{(n)}, \mathcal{M}_{1:K})\right) \right]$$

$$+ \mathrm{KL}\left(q_\phi\left(\mathcal{M}_{1:K}\right) \,\|\, p\left(\mathcal{M}_{1:K} \mid X\right)\right).$$

# B  ADDITIONAL EXPERIMENTAL RESULTS

## B.1  ABLATION STUDIES

**Effect of number of samples per component.** We investigate the effect of a decreasing number of samples per mixture component on the performance of MCD, as the number of ground truth SCMs $K^*$ increases. We consider synthetic data of dimension $D = 10$, and run MCD, Rhino, PCMCI$^+$ and Rhino (grouped) on $N = 1000$ samples generated from $K^*$ SCMs for increasing values of $K^*$, with $K = 2K^*$. The results are presented in Figure 6. MCD suffers a gradual decrease in model performance, with roughly a $25\%$ decrease in F1 and $11\%$ decrease in AUROC from $K^* = 1$ to $K^* = 100$. Meanwhile, the performance of Rhino falls off more drastically and becomes equivalent to random guessing for large $K^*$. The performance of PCMCI$^+$ (grouped) also decreases quite rapidly with the increase in $K^*$. Notably, MCD maintains a higher level of performance than Rhino (grouped), possibly due to the weight-sharing scheme.

**Using ground truth membership assignments.** We assess MCD performance with learned versus ground-truth membership associations on synthetic data with $D = 10$. As before, we set $K = 2K^*$. Figure 7 shows the results of this ablative experiment. The performance of MCD with ground truth

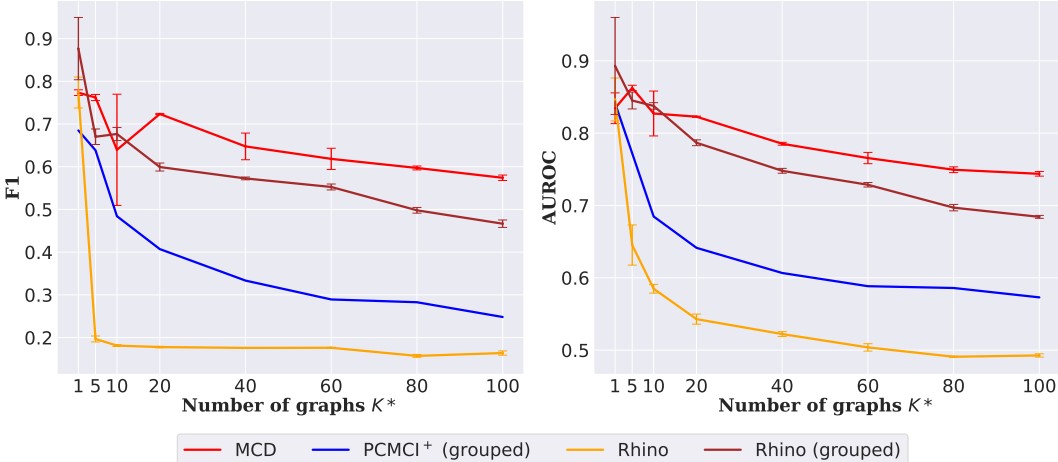

Figure 6: Effect of increasing number $K^*$ of underlying SCMs on F1 and AUROC on synthetic data with $D = 10$. MCD's performance declines gradually with decreased number of samples per mixture component, while Rhino's performance decays drastically. The performance of Rhino (grouped) decays slightly faster than MCD.

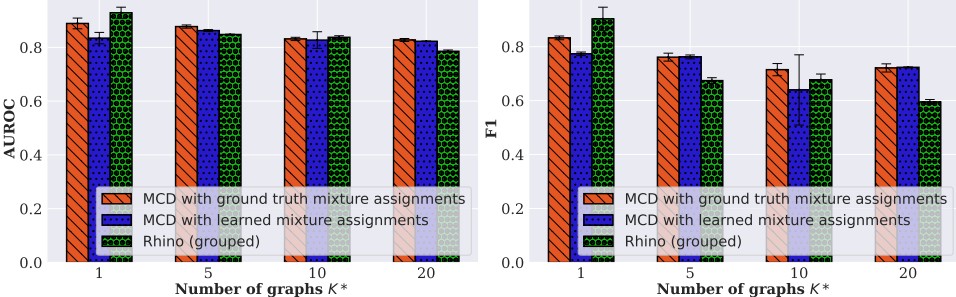

Figure 7: Comparison of model performance of MCD with ground-truth versus learned mixture assignments and Rhino (grouped) on synthetic data with $D = 10$. Expectedly, MCD performs better with explicit information about the cluster assignments, but it achieves comparable performance even with learned membership information.

labels is theoretically an upper bound on its performance. Encouragingly, we observe that our model performs very close to this upper bound (barring a few anomalous seeds). The largest difference in performance is observed for $K^* = 1$, where MCD learns two separate causal models to explain a single mode. We also observe that for $K^* > 1$, Rhino (grouped) performs a similar or slightly worse level of performance than MCD. This is due to the reduced number of samples per graph, which MCD is more robust to due to weight sharing (as implemented in Equation 10).

**Robustness of MCD to the misspecification of number of models.** We examine the performance of MCD when the number of mixture components $K$ is misspecified, and does not equal the true number of underlying components $K^*$. Figure 8 shows the performance of our model as a function of $K$ for synthetic data with dimensionality $D = 10$ and ground truth number of graphs $K^* = 10$. We note that when the number of models is underspecified, our model performs poorly as expected since it cannot fully explain all the modes in the data. Surprisingly, the performance increases with increasing $K$. The clustering accuracy and performance metrics show high standard deviation when $K$ is set to the true number of mixture components $K^* = 10$. While some runs achieve high clustering accuracy, others tend to saturate at a suboptimal grouping when $K = K^*$. On the other hand, when $K > K^*$, the additional SCMs are used as 'buffers' and the correct grouping is learned during the later epochs as the SCMs are inferred more accurately. This phenomenon is further explored in Appendix B.3

## B.2    CLUSTERING ACCURACY FOR $D = 5, 10, 20$ ON SYNTHETIC DATASETS

Figure 9 shows the clustering accuracy for different values of $D$ on the synthetic datasets. For all settings, we set the hyperparameter $K = 2K^*$. We observe that for all values of $K^* > 1$, the

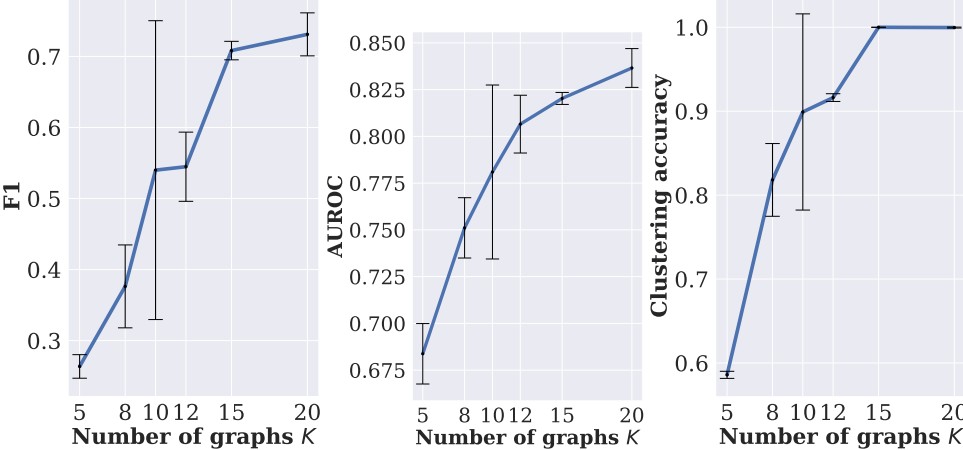

Figure 8: Performance of MCD as a function of hyperparameter input $K$ on synthetic data with $D = 10, K^* = 10$. Surprisingly, MCD performs better when the number of graphs is overspecified.

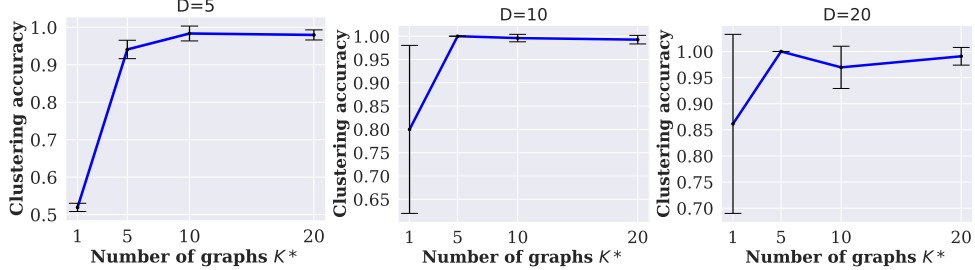

Figure 9: Plot showing the clustering accuracy vs $K^*$ on the synthetic datasets for $D = 5, 10, 20$. We observe that for $K^* > 1$, the clustering accuracy is high ($> 95\%$) on average.

clustering accuracy is, on average, above $95\%$, while it remains low for $K^* = 1$. As noted earlier, the low clustering accuracy for $K^* = 1$ is expected since the single mode in the data distribution is 'split' across two learnt causal graphs.

### B.3 CLUSTERING PROGRESSION WITH TRAINING

We analyze the progression of clustering accuracy and the number of unique graphs learned with the number of training steps. As training progresses, not all $K$ graphs are utilized. We count only those graphs for which there exists at least one associated sample. Figure 10 shows the plots. We observe that when $K = 20$, as training progresses, the algorithm groups together points from different causal graphs until they converge to the "true" number of causal graphs $K^* = 10$ and clustering accuracy converges to (approximately) $100\%$; however, when $K = 10$, we observe that the number of unique graphs can sometimes fall below $K^* = 10$, resulting in a poor clustering accuracy.

### B.4 NETSIM VISUALIZATION

Figure 11 shows a visualization of a heatmap of the predictions for the Netsim-permuted dataset. The 3 ground truth adjacency matrices and the top-3 discovered adjacency matrices, ranked by the frequency of prediction, are shown. All 3 matrices achieve a high AUROC score, even though the poor calibration of scores results in the prediction of many spurious edges.

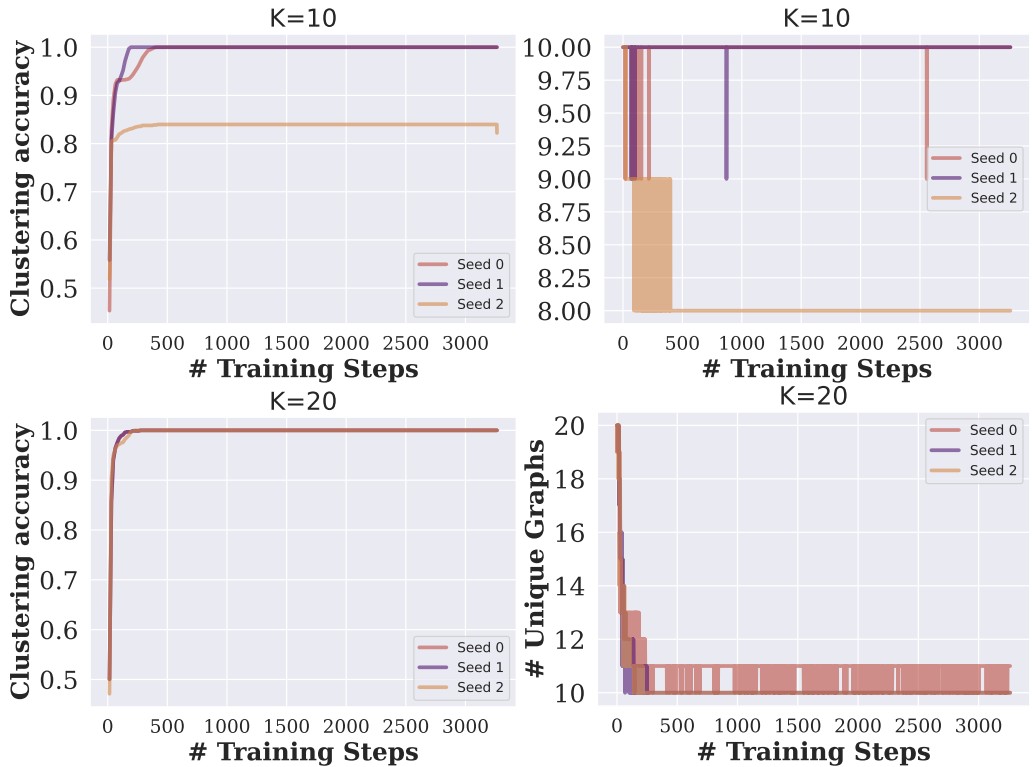

Figure 10: Plots showing the progression of (left) clustering accuracy (right) number of unique learned graphs with the number of training steps on the synthetic dataset with $D = 10, K^* = 10$. We observe that as training progresses, clustering accuracy increases for both the $K = 10$ and $K = 20$ runs; however when $K = 10$, some runs tend to learn a lower number of

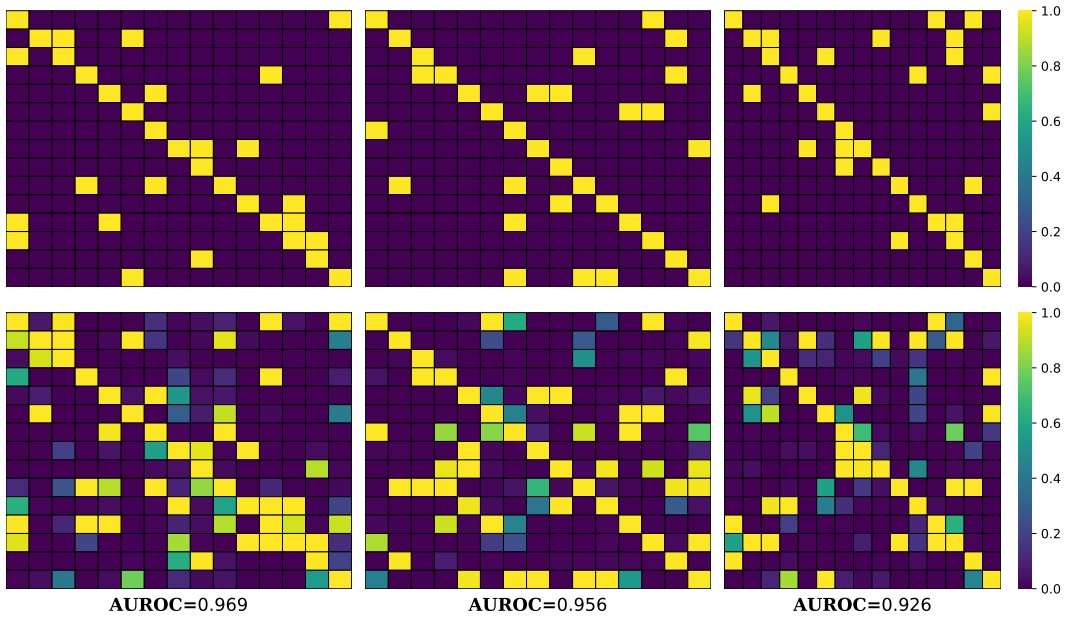

Figure 11: Heatmap for the Netsim-Permuted dataset showing the (top) adjacency matrices of the ground-truth causal graphs, and the (bottom) edge probabilities for the top-3 discovered adjacency matrices (ranked by frequency of occurrence). We also report the graph-wise AUROC metrics.

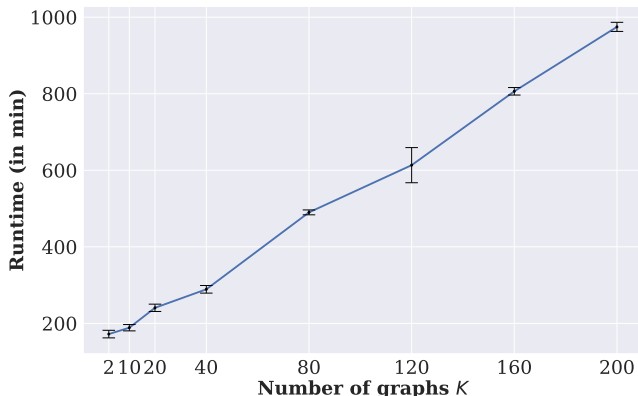

Figure 12: Run time plot of MCD as a function of $K$. A $100\times$ increase in $K$ from 2 to 200 results in a less than $10\times$ increase in run-time.

### B.5 TIMING ANALYSIS

In this section, we analyze the run-time of MCD as a function of the hyperparameter $K$. As noted in Section 3.2, MCD, in theory, needs roughly $K$ times more operations than Rhino in each epoch due to the evaluation of the expectation over the variational distribution $r_\psi \left( Z^{(n)} \mid X^{(n)} \right)$ while calculating the ELBO. However, we show that, in practice, this does not translate to a $K$ times increase in model runtime. We measure and plot the total runtime for training our model for the synthetic dataset with $D = 10$ nodes as a function of $K$. Figure 12 shows the plot.

We observe that although the plot shows an approximately linear trend, the slope is much lesser than 1. In fact, a $100\times$ increase in $K$ from 2 to 200 results in a less than $10\times$ increase in run-time. Thus, MCD scales reasonably well with the number of mixture components $K$.

## C IMPLEMENTATION DETAILS

In this section, we describe how we model the terms in Equation 3. We follow the implementation described in Rhino (Gong et al., 2022) due to its ability to model instantaneous effects and history-dependent noise. Similar to Rhino, we make a simplifying assumption about the functional form: we assume that we have an additive noise model since it is known to be identifiable (Zhang et al., 2015). Under causal model $\mathcal{M}_k = (\mathcal{G}_k, \theta_k)$,

$$X_t^{i,(n)} = f_k^i(\mathrm{Pa}_{\mathcal{G}_k}^i(< t), \mathrm{Pa}_{\mathcal{G}_k}^i(t)) + \epsilon_t^i. \tag{7}$$

We model the above equation as follows:

$$X_t^{i,(n)} = f_k^i(\mathrm{Pa}_{\mathcal{G}_k}^i(< t), \mathrm{Pa}_{\mathcal{G}_k}^i(t)) + g_k^i(\mathrm{Pa}_{\mathcal{G}_k}^i(< t), \epsilon_t^i).$$

where the function $f_k^i$ models the functional relationship between the nodes and $g_k^i$ models the history dependence of the exogenous noise for node $i$ under causal model $k$. The noise variables $\epsilon_t^i$ are described using a conditional spline flow model, akin to (Gong et al., 2022).

$$p_{g_k^i}(g_k^i(\epsilon_t^i) \mid \mathrm{Pa}_{\mathcal{G}_k}^i(< t)) = p_\epsilon(\epsilon_t^i) \left| \frac{\partial (g_k^i)^{-1}}{\partial \epsilon_t^i} \right| \tag{8}$$

with $\epsilon_t^i$ modeled as independent Gaussian noise.

The marginal likelihood under each model $\mathcal{M}_k$ can be further simplified as follows, using the causal Markov assumption:

$$\log p_\theta \left( X_{1:T}^{(n)} \middle| \mathcal{M}_{Z^{(n)}} \right) = \sum_{t=L}^{T} \sum_{i=1}^{D} \log p_\theta \left( X_t^{i,(n)} \middle| \mathrm{Pa}_{\mathcal{G}_{Z^{(n)}}}^i(< t), \mathrm{Pa}_{\mathcal{G}_{Z^{(n)}}}^i(t) \right)$$

$$= \sum_{t=L}^{T} \sum_{i=1}^{D} \log p_{g_{Z^{(n)}}^i} \left( z_t^{i,(n)} \middle| \mathrm{Pa}_{\mathcal{G}_{Z^{(n)}}}^i(< t) \right) \tag{9}$$

where $z_t^{i,(n)} = X_t^{i,(n)} - f_{Z^{(n)}}^i \left( \mathrm{Pa}_{\mathcal{G}_{Z^{(n)}}}^i(< t), \mathrm{Pa}_{\mathcal{G}_{Z^{(n)}}} \right)$.

The functional relationships are implemented using neural networks $\xi$ and $\ell$ in the following equation:

$$f_{Z^{(n)}}^i \left( \text{Pa}_{\mathcal{G}_{Z^{(n)}}}^i (< t), \text{Pa}_{\mathcal{G}_{Z^{(n)}}}^i (t) \right) = \xi \left( \left[ \sum_{\tau=0}^{L} \sum_{j=1}^{D} (\mathcal{G}_{Z^{(n)}})_\tau^{ji,(n)} \, \ell \left( \left[ X_{t-\tau}^{j,(n)}, (\theta_{Z^{(n)}})_\tau^{j,(n)} \right] \right), (\theta_{Z^{(n)}})_0^{i,(n)} \right] \right) \tag{10}$$

where $\theta_{Z^{(n)}}$ are embeddings corresponding to model $Z^{(n)}$, and $\xi$ and $\ell$ are multi-layer perceptron networks that are shared across all causal models $\mathcal{M}_{1:K}$. A similar architecture is used for the hypernetwork that predicts parameters for the conditional spline flow model.

The prior distribution $p(\mathcal{M}_{1:K})$ is modeled as follows:

$$p_\theta(\mathcal{M}_{1:K}) \propto \prod_{k=1}^{K} \exp \left( -\lambda \left\| (\mathcal{G}_k)_{1:T} \right\|^2 - \sigma h \left( (\mathcal{G}_k)_0 \right) \right). \tag{11}$$

The first term is a sparsity constraint and $h\left( (\mathcal{G}_k)_0 \right)$ is the acyclicity constraint from (Zheng et al., 2018). The adjacency matrix $\mathcal{G}_i$ is represented as a product of independent Bernoulli distributions.

### C.1 CALCULATION OF CLUSTERING ACCURACY

We would like to evaluate the accuracy of our method in grouping samples based on the underlying SCMs. However, the assigned cluster indices by the model and the 'ground-truth' cluster indices might not match nominally, even though they refer to the same grouping assignment. For example, the cluster assignment of $(1, 1, 1, 2, 2)$ for $N = 5$ points is equivalent to the assignment $(2, 2, 2, 1, 1)$. In other words, we want a permutation invariant accuracy metric between the inferred cluster assignments $\tilde{Z}$ and true cluster assignments $Z$ with $\tilde{Z}, Z \in \mathbb{N}^N$. We define

$$\text{Cluster Acc.} \left( \tilde{Z}, Z \right) = \max_{\pi \in S_K} \frac{1}{N} \sum_{n=1}^{N} 1 \left( \pi(\tilde{Z}_i) = Z_i \right)$$

with $S_K$ denoting the permutation group over $K$ elements. Evaluating the cluster accuracy naively would require $K!$ operations. However, we use the Hungarian algorithm to find the correct permutation in $O(K^3)$ time[1].

## D EXPERIMENTAL DETAILS

### D.1 HYPERPARAMETER DETAILS

For all our experiments with MCD, we set the lag value $L = 2$ for all the considered methods. The coefficient of the DAG penalty term in the loss function was set to $1 + n_e$, where $n_e$ is the epoch number. We used the rational spline flow model described in (Durkan et al., 2019). For all our experiments, we use the linear rational spline flow model, with 8 bins. The MLPs $\ell$ and $\xi$ have 2 hidden layers each and with LeakyReLU activation functions. Other hyperparameters used for training are summarized in Table 3. We used embedding dimension $e = 128$ for all our experiments.

| Dataset | Matrix LR | Likelihood LR | Batch size | Matrix temperature | Num. Epochs |
|---|---|---|---|---|---|
| Synthetic ($D = 5$) | $10^{-2}$ | $10^{-3}$ | 128 | 1 | 200 |
| Synthetic ($D = 10$) | $10^{-2}$ | $10^{-3}$ | 128 | 1 | 200 |
| Synthetic ($D = 20$) | $10^{-2}$ | $10^{-3}$ | 128 | 1 | 200 |
| Netsim | $10^{-2}$ | $10^{-3}$ | 32 | 0.25 | 200 |
| Netsim-Permuted | $10^{-2}$ | $10^{-3}$ | 32 | 0.25 | 200 |
| DREAM3 | $10^{-2}$ | $10^{-3}$ | 8 | 0.25 | 500 |

Table 3: Table showing the hyperparameters for MCD on different datasets.

**Baselines.** Rhino was trained with similar hyperparameters as MCD on all datasets except Netsim, on which it was trained for 400 epochs with learning rates $10^{-4}$ for the likelihood and $10^{-3}$ for the graph logits. As before, we used the linear rational spline flow model with 8 bins. For all other baselines, the default hyperparameter values are used. For Rhino and MCD, which parameterize the causal graphs as Bernoulli distributions over each edge, we use the inferred edge probability matrix

---

[1] This approach and implementation are adapted from `https://smorbieu.gitlab.io/accuracy-from-classification-to-clustering-evaluation/`

as the "score", and evaluate the AUROC metric between the score matrix and the true adjacency matrix. For DYNOTEARS, we use the absolute value of the output scores and evaluate the AUROC. For PCMCI+ and VARLiNGAM, since they only output adjacency matrices, we directly evaluate the AUROC between the predicted and true adjacency matrices.

## D.2 POST-PROCESSING THE OUTPUT OF PCMCI$^+$

PCMCI$^+$ produce Markov equivalence classes rather than fully oriented causal graphs. To make its outputs comparable, we post-process the resultant edges. We symmetrically set both the entries corresponding to a bidirectional edge to 1 in the adjacency matrix, and ignore the edges (i.e., set the corresponding entries in the adjacency matrix to 0) whose orientations are undecided.

## D.3 EVALUATION ON NETSIM AND DREAM3 DATASETS

The Netsim and DREAM3 datasets used in the evaluation provide ground-truth time-aggregated causal graphs. In order to make our model output comparable, we follow the procedure outlined in (Gong et al., 2022) to convert the time-lag adjacency matrix to an aggregated matrix. The $(i, j)^{\text{th}}$ entry of the aggregated matrix $\mathcal{G}_{\text{agg}}$ is 1 iff $\mathcal{G}_\ell^{ij} = 1$ for some lag value $\ell$ in the time-lag matrix $\mathcal{G}$. Both Rhino and MCD represent the edges as Bernoulli random variables and hence output a probability score for each edge. For evaluating the F1 score of Rhino and Netsim, we threshold the probability values at 0.5, i.e., edges with a probability $\geq 0.5$ are considered as predicted edges.

## D.4 PAIR-WISE GRAPH DISTANCE IN THE MIXTURE DISTRIBUTIONS

Table 4 shows the pairwise graph distances between the ground-truth graphs of the mixture distributions used in the paper. We calculate the Structural Hamming Distance (SHD) between every pair of graphs in the mixture, and report the mean, standard deviation, minimum and maximum values.

| Dataset | $D$ | $K^*$ | Mean SHD | Std. Dev. SHD | Min. SHD | Max. SHD |
|---|---|---|---|---|---|---|
| Synthetic | 5 | 5 | 24.00 | 1.95 | 20 | 27 |
| Synthetic | 5 | 10 | 23.24 | 3.37 | 14 | 30 |
| Synthetic | 5 | 20 | 22.61 | 3.29 | 13 | 31 |
| Synthetic | 10 | 5 | 53.40 | 3.83 | 48 | 59 |
| Synthetic | 10 | 10 | 54.09 | 3.26 | 45 | 61 |
| Synthetic | 10 | 20 | 54.27 | 3.73 | 44 | 64 |
| Synthetic | 20 | 5 | 113.80 | 5.62 | 101 | 120 |
| Synthetic | 20 | 10 | 111.91 | 5.42 | 99 | 123 |
| Synthetic | 20 | 20 | 113.59 | 5.06 | 98 | 124 |
| DREAM3 | 100 | 5 | 517.60 | 202.13 | 234 | 896 |
| Netsim | 5 | 14 | 2.59 | 1.17 | 1 | 5 |
| Netsim-permuted | 15 | 3 | 34.00 | 1.63 | 32 | 36 |

Table 4: Pair-wise graph statistics for experimental datasets used in the paper.

## E TOY EXAMPLE

Consider a dataset where each sample $X^{(n)}$ from the dataset $\left\{ X_{1:T}^{1:D,(n)} \right\}_{n=1}^N$ is generated from one out of the two following SCMs with equal probability:

$$X_t^{1,(n)} = 0.4 X_{t-1}^{2,(n)} + 0.6 X_t^{3,(n)} + \epsilon_1^{(n)}$$
$$X_t^{2,(n)} = 0.3 X_{t-1}^{3,(n)} + 0.3 X_t^{3,(n)} + \epsilon_2^{(n)}$$
$$X_t^{3,(n)} = 0.5 X_{t-1}^{1,(n)} + \epsilon_3^{(n)}$$

(or)

$$X_t^{1,(n)} = 0.7 X_{t-1}^{3,(n)} - 0.2 X_t^{2,(n)} + \epsilon_1^{(n)}$$
$$X_t^{2,(n)} = 0.2 X_{t-1}^{1,(n)} + 0.4 X_t^{3,(n)} + \epsilon_2^{(n)}$$
$$X_t^{3,(n)} = -0.3 X_{t-1}^{1,(n)} + \epsilon_3^{(n)}.$$

These SCMs can be represented through the temporal causal graphs given in Figure 13.

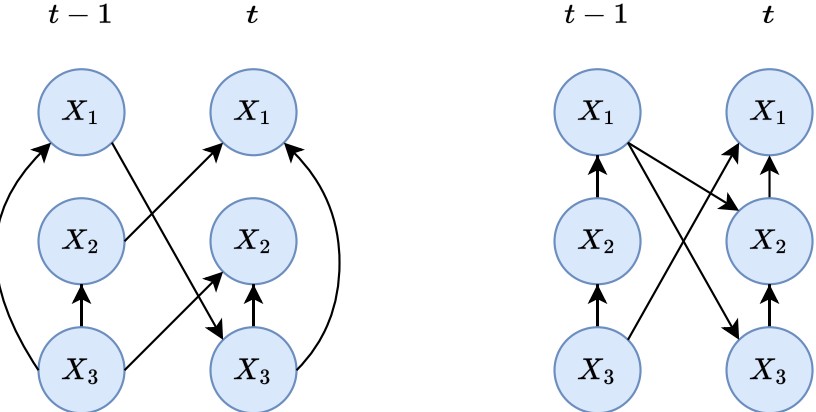

Figure 13: Temporal causal graphs which represent the causal relationships encoded by the SCMs.

However, if the graph membership of the samples is unknown, inferring a single causal graph to explain the causal relationships from the dataset would result in spurious causal relationships. For example, going by conditional independence tests, note that none of the nodes would be conditionally independent of each other for any conditioning set. This is also exemplified in the output of the PCMCI$^+$ algorithm, where a fully connected graph is inferred as shown in Figure 14. Thus, it is crucial to use a mixture distribution to model observational data coming from such heterogeneous data distributions.

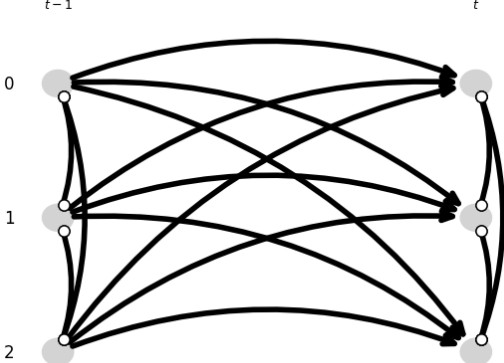

Figure 14: PCMCI$^+$ output on the toy-example. The algorithm infers a fully connected graph with many spurious causal relationships.

