# OpenReview forum: "Discovering Mixtures of Structural Causal Models from Time Series Data"
_ICLR.cc/2024/Conference — Submitted to ICLR 2024_

### Official Review · Reviewer_bLPE · 2023-10-30

**Soundness:** 3 good
**Presentation:** 2 fair
**Contribution:** 2 fair
**Rating:** 5
**Confidence:** 3

**Summary:**

The authors proposed an approach to perform causal discovery from time series data originating from mixtures of different causal models. This approach can simultaneously infer the underlying causal graphs and the posterior likelihood of each sample belonging to a specific mixture component by maximizing the evidence lower bound, on top of the framework of the Rhino algorithm.

**Strengths:**

1. The problem setting is interesting, and the approach holds potential for broad applicability.
2. The authors conduct extensive experiments on simulated and two real-world datasets, compared with several baselines.
3. The authors characterize a sufficient condition for the identifiability of such mixture models and explain the relationship between the constructed evidence lower bound and the data likelihood.
4. The authors provide ablation studies.

**Weaknesses:**

1. Certain notations and explanations remain unclear, and these will be described in the upcoming Questions section.
2. Some details concerning the comparison between the proposed method and baseline algorithms in the experiments are perplexing, and these questions will also be raised in the later section.
3. The baselines used in the paper are not algorithms designed for multiple DAGs and mixture SCMs. Even though the authors mentioned other algorithms designed for multiple DAGs, none has been applied in the comparison.

**Questions:**

**1. Notations and explanations**

1.1 The SCM described in equation (1) differs from equation (6) and also the equation below equation (6); which one is correct? Do you assume additive noise?

1.2 In Theorem 1 equation (*), what is the definition of $a_i$? Does it mean one sample from $i$th SCM?

1.3 In Theorem 1, what are $g_1$ and $g_2$? Should they be $h_1$ and $h_2$?

1.4 As mentioned in the main paper, "This highlights the idea that learning a mixture model is only beneficial when the underlying SCMs differ from one another significantly. " could you clarify what the significant difference is here?

**2. Details in the experiment**

2.1 In the experiment section, "(per sample) indicates that the baseline predicts one graph per sample." how to apply the algorithm with only one sample? For example, PCMCI$^{+}$ needs CI tests, and CI tests need a set of samples instead of only one sample.

2.2 In the experiment performance section, AUROC and F1 are used as metrics. However, it is not mentioned which kind of AUROC and F1 refer to adjacency or orientation AUROC/F1?

2.3 As far as I know, the Rhino algorithm is not designed for heterogeneous time series data. In the experimental section, would it be feasible to compare the proposed algorithm with "Rhino (grouped)" as well? This would involve applying the Rhino algorithm to the grouped data based on the true underlying causal graph. Considering this, the results regarding Rhino in Figures 3 and 6 could be more comprehensive if grouped data were utilized.

2.4 Could the post-processing of PCMCI$^{+}$ outputs potentially affect the AUROC/F1 scores of PCMCI$^{+}$? If it does, would this influence tend to favor the proposed method in the comparison results?

2.5 In the synthetic data experiment, the metrics are averaged across 3 runs. Does each run requires a newly generated synthetic dataset? Personally, 3 runs seem to be a limited number for comprehensive evaluation.

2.6 In Figure 3, it seems a little strange that the performance of PCMCI$^{+}$ (grouped) is better with $D=10$ than with $D=5$. The same phenomenon happened in other algorithms. Do you have any clue on why this happened?

2.7 In Figure 4, "The accuracy is averaged across 3 runs and across data dimensionality D = 5, 10, 20.", is it more appropriate to plot the results for $D=5,10,20$ separately as the value of $D$ could affect the accuracy? Again, 3 runs seem limited to me.

2.8 Could you explain more about "The clustering accuracy and performance metrics show high standard deviation when K is set to the true number of mixture components K* = 10." which is stated in the appendix and explain more about the "buffers" when $K>K^{*}$?

**3. Other questions**

3.1 Why maximizing ELBO is equivalent to maximizing or minimizing each term separately in the log-likelihood in terms of $\theta, \phi$, and $\psi$ as $\phi$ and $\psi$ appear together in the second term?

3.2 Is it better to mention the Rhino algorithm in the related work section and briefly explain how it works as the proposed algorithm is built on top of Rhino?

3.3 Is it essential to verify whether the data satisfies condition (*) to ensure the reliability of the results? If so, how to verify this?

3.4 The paper mentions that people can also infer functional equations through the proposed model. Could you help me locate the related outputs in the experiment section? Does this mean the causal effect?

3.5 Has any code been provided?

---

> ### Author Response · Authors · 2023-11-21
>
> We thank the reviewer for their useful feedback and questions. Below we address the main concerns:
>
> # Weaknesses:
>
> > 3. The baselines used in the paper are not algorithms designed for multiple DAGs and mixture SCMs. Even though the authors mentioned other algorithms designed for multiple DAGs, none has been applied in the comparison.
>
> We are not aware of any baselines dealing with time-series data that are designed for multiple DAGs. All the algorithms mentioned in the Related Work section are designed for independent data, and hence not directly applicable to our setting. We would appreciate any concrete pointers to comparable algorithms.
>
> # Questions
> 1. Notations and explanations
> > 1.1 The SCM described in equation (1) differs from equation (6) and also the equation below equation (6); which one is correct? Do you assume additive noise?
>
> We presented equation 1 for preliminaries, however we assume additive noise in our work due to its structural identifiability. We have clarified this in the updated draft, in Section 3.
>
> > 1.2 In Theorem 1 equation (*), what is the definition of $a_i$? Does it mean one sample from $i$th SCM?
>
> $a_i$ refers to any point in the support of the mixture distribution such that the condition (*) is satisfied (as specified in mathematical notation in the theorem statement). It is not a ‘sample’ from the ith SCM in the sense that it is not randomly sampled from the SCM, but it can be potentially explicitly chosen for the condition to hold. We have added this clarification to the draft in Appendix A.3.
>
> > 1.3 In Theorem 1, what are $g_1$ and $g_2$? Should they be $h_1$ and $h_2$?
>
> Thank you for pointing out the typo. Yes, they should be $h_1$ and $h_2$. We have corrected this in the new draft in the theorem statement.
>
> > 1.4 As mentioned in the main paper, "This highlights the idea that learning a mixture model is only beneficial when the underlying SCMs differ from one another significantly. " could you clarify what the significant difference is here?
>
> By “significant difference”, we mean that the causal graphs in the mixture distribution differ from one another significantly in terms of their edges and resulting data likelihood.  This is also exemplified by the following table of pair-wise SHDs between the constituent graphs of the mixture models.
>
> | Dataset         | D  | K* | Mean SHD | Std Dev SHD | Min SHD | Max SHD |
> |-----------------|----|----|----------|-------------|---------|---------|
> | Netsim          |  5 | 14 |     2.59 |        1.17 |       1 |       5 |
> | Netsim-permuted | 15 |  3 |    34.00 |        1.63 |      32 |      36 |
>
> We note a notably higher clustering accuracy and improved performance in causal discovery when evaluating the Netsim-permuted dataset. This is in contrast to the Netsim dataset, where the pairwise Structural Hamming Distance (SHD) is smaller, indicating that the mixture graphs in this dataset are more similar to each other.
>
>
> 2. Details in the experiment
> > 2.1 In the experiment section, "(per sample) indicates that the baseline predicts one graph per sample." how to apply the algorithm with only one sample? For example, PCMCI+ needs CI tests, and CI tests need a set of samples instead of only one sample.
>
> Each sample is a time-series of length $T$, while PCMCI+ performs conditional independence testing over time-windows of length $L$, where $T > L$. Thus, CI tests are conducted on a set of time windows per time series.
>
> > 2.2 In the experiment performance section, AUROC and F1 are used as metrics. However, it is not mentioned which kind of AUROC and F1 refer to adjacency or orientation AUROC/F1?
>
> Thanks for asking for the clarification. All the reported metrics are orientation AUROC and F1. We have updated the Experiments section of the draft to clarify this as well.
>
>
> > 2.3 As far as I know, the Rhino algorithm is not designed for heterogeneous time series data. In the experimental section, would it be feasible to compare the proposed algorithm with "Rhino (grouped)" as well? This would involve applying the Rhino algorithm to the grouped data based on the true underlying causal graph. Considering this, the results regarding Rhino in Figures 3 and 6 could be more comprehensive if grouped data were utilized.
>
> You are correct that Rhino is not designed for heterogeneous time series data. Based on your suggestion, we have added Rhino (grouped) results on the synthetic datasets in Figures 3, 6 and 7. We observe that Rhino (grouped) on average performs at the same level or worse than MCD on the synthetic datasets (especially for higher number of graphs). This is due to the reduced number of samples per graph, which MCD is more robust to due to weight sharing (as implemented in Equation 10).

---

> > ### Author Response · Authors · 2023-11-21
> >
> > > 2.4 Could the post-processing of PCMCI+ outputs potentially affect the AUROC/F1 scores of PCMCI+? If it does, would this influence tend to favor the proposed method in the comparison results?
> >
> > Post-processing of PCMCI+ outputs actually favors PCMCI+ instead of the proposed method. We chose this for a more fair comparison.  PCMCI+ can only output a Markov Equivalence Class (MEC) for the instantaneous matrix, and hence the orientation of many edges are undetermined.
> >
> > We also compared this post-processing scheme to the one presented in [1], where upto 3000 possible DAGs are enumerated from the MEC. The results on the synthetic datasets with $D=5, 10$ are presented below:
> >
> > Orientation F1 scores with enumeration:
> > | **D** | **K** | **PCMCI+ (per sample)** | **PCMCI+ (grouped)** | **PCMCI+ (single graph)** |
> > |:-----:|:-----:|:-----------------------:|:--------------------:|:-------------------------:|
> > |   5   |   1   |          0.094          |         0.454        |           0.454           |
> > |   5   |   5   |          0.095          |         0.494        |           0.242           |
> > |   5   |   10  |          0.082          |         0.521        |           0.146           |
> > |   5   |   20  |          0.086          |         0.383        |           0.182           |
> > |   10  |   1   |          0.065          |         0.621        |           0.621           |
> > |   10  |   5   |          0.074          |         0.583        |           0.176           |
> > |   10  |   10  |          0.083          |         0.407        |           0.096           |
> > |   10  |   20  |          0.063          |         0.336        |           0.108           |
> >
> > Orientation F1 scores with our post-processing scheme:
> > | **D** | **K** | **PCMCI+ (per sample)** | **PCMCI+ (grouped)** | **PCMCI+ (single graph)** |
> > |:-----:|:-----:|:-----------------------:|:--------------------:|:-------------------------:|
> > |   5   |   1   |          0.098          |         0.600        |           0.600           |
> > |   5   |   5   |          0.100          |         0.534        |           0.310           |
> > |   5   |   10  |          0.086          |         0.574        |           0.266           |
> > |   5   |   20  |          0.102          |         0.477        |           0.210           |
> > |   10  |   1   |          0.070          |         0.685        |           0.685           |
> > |   10  |   5   |          0.092          |         0.638        |           0.205           |
> > |   10  |   10  |          0.093          |         0.484        |           0.151           |
> > |   10  |   20  |          0.073          |         0.407        |           0.104           |
> >
> > We observe that PCMCI+ seems to evaluate better with our post-processing scheme compared to the one presented in [1].
> >
> > > 2.5 In the synthetic data experiment, the metrics are averaged across 3 runs. Does each run requires a newly generated synthetic dataset? Personally, 3 runs seem to be a limited number for comprehensive evaluation.
> >
> > Based on your suggestion, we have re-run all experiments on the synthetic dataset from the main paper for 5 runs instead of 3, and updated Figure 3 in the draft to reflect this change. We do not observe significant changes when we use 5 runs vs 3. We run 5 runs on the same dataset, i.e. we do not use a newly generated synthetic dataset for each run.
> >
> > > 2.6 In Figure 3, it seems a little strange that the performance of PCMCI+  (grouped) is better with D=10  than with D=5. The same phenomenon happened in other algorithms. Do you have any clue on why this happened?
> >
> > Since the synthetic datasets are randomly generated, some settings (e.g. D=10) could be inherently more difficult than the others (e.g. D=5) simply due to the stochasticity of the generation process. Our hypothesis seems to be supported by the observation that all methods have similar trends across the two datasets.
> >
> >
> > > 2.7 In Figure 4, "The accuracy is averaged across 3 runs and across data dimensionality D = 5, 10, 20.", is it more appropriate to plot the results for D=5,10,20 separately as the value of D could affect the accuracy? Again, 3 runs seem limited to me.
> >
> > Based on your suggestion, we have reported results by averaging over 5 runs, and also added plotted the results for $D=5, 10$ and $20$ separately. The new plots are in Appendix B.2. We observe a similar trend to the one we observed earlier, when averaged across all data dimensionalities, with clustering accuracy remaining above $95\%$ for $K^\ast>1$.

---

> ### Author Response · Authors · 2023-11-21
>
> > 2.8 Could you explain more about "The clustering accuracy and performance metrics show high standard deviation when K is set to the true number of mixture components K* = 10." which is stated in the appendix and explain more about the "buffers" when K>K*?
>
> We have added a new section Appendix B.3 with additional experiments for the $D=10, K*=10$ synthetic dataset with $K=10$ and $K=20$ to illustrate our assertion. We plot both the number of unique graphs and the clustering accuracy as a function of number of training steps. We observe that when $K=20$, as training progresses, the algorithm groups together points from different causal graphs until they more or less converge to the “true” number of causal graphs and clustering accuracy converges to (approximately) 100%; however, when $K=10$, we observe that for some runs, the number of unique graphs can sometimes fall below $K^\ast=10$, and thus the clustering accuracy can suffer.
>
> 3. Other questions
> > 3.1 Why maximizing ELBO is equivalent to maximizing or minimizing each term separately in the log-likelihood in terms of $\theta$, $\phi$, and $\psi$ as $\phi$ and $\psi$ appear together in the second term?
>
> We did not mean that each term in the log-likelihood is maximized or minimized separately; indeed we mean the joint optimization of the ELBO in terms of $\theta, \phi$ and $\psi$ is equivalent to optimizing the log-likelihood. We have updated Section 4 in the draft to clarify this.
>
> > 3.2 Is it better to mention the Rhino algorithm in the related work section and briefly explain how it works as the proposed algorithm is built on top of Rhino?
>
> Thank you for the suggestion! we have already included a self-sufficient description of the Rhino algorithm in Appendix C due to page limit.
>
> > 3.3 Is it essential to verify whether the data satisfies condition (\*) to ensure the reliability of the results? If so, how to verify this?
>
> Yes, it is essential to ensure the identifiability of the underlying mixture model in order to ensure that the problem of causal discovery is a statistically feasible problem. Condition (\*) gives one sufficient condition to verify this. While condition (\*) can be difficult to verify exactly in most scenarios since the log-likelihood is rarely explicitly known, in practice one can “approximately” check it given good enough approximations to the log-likelihood functions of the underlying causal models, by explicitly checking if any of the samples satisfy the condition.
>
> > 3.4 The paper mentions that people can also infer functional equations through the proposed model. Could you help me locate the related outputs in the experiment section? Does this mean the causal effect?
>
> We do not benchmark the causal effect, since our focus in this paper is on causal discovery. We will work on benchmarking the causal effect in a future iteration of the paper.
>
> > 3.5 Has any code been provided?
>
> We will release our code after the peer review process.
>
> References:
> [1] Gong, Wenbo, et al. "Rhino: Deep causal temporal relationship learning with history-dependent noise." arXiv preprint arXiv:2210.14706 (2022).

---

> > ### Comment · Reviewer_bLPE · 2023-11-22
> >
> > Thank you for the extensive classification, additional experiments, and visualizations provided. While most of my queries have been answered, there are still a couple of suggestion/questions:
> >
> > 1.1 If the SCM assumes additive noise, perhaps mentioning it in the abstract or contribution section would be beneficial.
> >
> > 2.1 Based on my understanding of PCMCI, the algorithm operates on a p-variate time series with length T, conducting CI tests across the entire set of time series samples rather than within a sample window. If my understanding is incorrect, could you direct me to the specific section in the PCMCI$^{+}$ paper discussing the time window $L$?
> >
> > 2.1' I have a follow-up question based on  2.1. Considering PCMCI$^{+}$ requires only one p-variate time series with length T (where the sample of time series needed is $n=1$), I'm curious about the input of PCMCI$^{+}$ in your experiments. Specifically, I had assumed that the input dimension for PCMCI$^{+}$ should be $p\times T$. How is a dataset sized $n\times p \times T$ fed into PCMCI$^{+}$, where $n$ signifies the number of time series samples?
> >
> > 2.6 Do you mean that some settings (e.g., D=10) could be inherent $\textit{easier}$ than others (e.g., D=5) simply due to the stochasticity of the generation process? I understand your point; just want to ask whether it is a typo in your answer.

---

> ### Author Response · Authors · 2023-11-23
>
> Thanks for your response. Below we answer your queries:
>
> > 1.1. If the SCM assumes additive noise, perhaps mentioning it in the abstract or contribution section would be beneficial.
>
> Thank you for the suggestion. We have clarified that we work with the additive noise assumption in the contributions section of the introduction.
>
> > 2.1 Based on my understanding of PCMCI, the algorithm operates on a p-variate time series with length T, conducting CI tests across the entire set of time series samples rather than within a sample window. If my understanding is incorrect, could you direct me to the specific section in the PCMCI$^+$ paper discussing the time window?
>
> Quoting from the PCMCI paper [1]
> "The nodes in a time series graph represent the variables at different lag times and a causal link $X^i_{t \rightarrow \tau} \rightarrow X_t^j$ exists if $X_{t-\tau}^i$ is not conditionally independent of $X_t^j$ given the past of all variables, formally defined by
> $X^i\_{t-\tau} \not\perp X\_t^j | (\mathbf{X}\_t^{-}- \\{ X\_{t-\tau}^i \\})$  with $\not\perp$ denoting the absence of a (conditional) independence, the vertical bar $|$ meaning 'conditional on', and $\mathbf{X}\_t^- - \\{ X_{t-\tau}^i\\} $ *denoting the past of all $N$ variables up to a maximum time lag $\tau_\text{max}$* excluding $X^i_{t-\tau}$."
> Since the PCMCI algorithm assumes causal stationary, the CI tests are carried out for time windows of length $\tau_\text{max}$, which denote the maximum time length for which the causal relationships are inferred.
>
>  > 2.1' I have a follow-up question based on 2.1. Considering PCMCI$^+$ requires only one p-variate time series with length T (where the sample of time series needed is $n=1$) I'm curious about the input of PCMCI$^+$ in your experiments. Specifically, I had assumed that the input dimension for PCMCI$^+$ should be $p \times T$. How is a dataset sized $n \times p \times T$ fed into PCMCI$^+$, where $n$ signifies the number of time series samples?
>
> PCMCI$^+$ as implemented in the Tigramite package, has two options based on whether there is a single time-series input of size $p \times T$ or multiple time-series of size $n \times p \times T$. When inferring a single graph per sample, we use the `analysis_mode='single'` setting. When inferring a graph per dataset we use the `analysis_mode='multiple'` option [2].
>
> > 2.6 Do you mean that some settings (e.g., D=10) could be inherently easier than others (e.g., D=5) simply due to the stochasticity of the generation process? I understand your point; just want to ask whether it is a typo in your answer.
>
> Yes, this is what we meant. Apologies for the typo in our response.
>
> References:
> [1] PCMCI: J. Runge, P. Nowack, M. Kretschmer, S. Flaxman, D. Sejdinovic, Detecting and quantifying causal associations in large nonlinear time series datasets. Sci. Adv. 5, eaau4996 (2019). https://advances.sciencemag.org/content/5/11/eaau4996
> [2] Tigramite package. https://jakobrunge.github.io/tigramite/#module-tigramite.data_processing

---

> > ### Comment · Reviewer_bLPE · 2023-11-23
> >
> > Thank you for the response.
> >
> > I understood your point regarding 2.1. What I intended to emphasize is in each CI test, all samples are used, whereas you're suggesting that the CI tests are established solely within certain ranges of pairs of random variables. We're discussing different aspects, and both interpretations hold validity in their respective contexts.

---

### Official Review · Reviewer_dKxB · 2023-10-30

**Soundness:** 1 poor
**Presentation:** 3 good
**Contribution:** 2 fair
**Rating:** 3
**Confidence:** 4

**Summary:**

An ELBO method is introduced for finding and assigning weight to component SEMs in a mixture and drawing causal inferences from this mixture.

**Strengths:**

The problem of identifying the different SCMs in a mixture model where each mixed dataset uses a different graph and parameterization is extremely important, so I’m glad it’s being addressed.

**Weaknesses:**

I do have some issues.

1.	If we turn to the experimental section of the paper, we get two examples: one for NetSim (which I’m very familiar with) and another for the DREAM3 gene network. Both of these have problems with respect to the goal of this paper, suggesting that the choice of experimental datasets could be improved.
2.	The problem with the NetSim data is that it’s not an extremely convincing time series, as the records in the simulation are spaced far enough apart in time to render the data nearly i.i.d. In fact, analyzing it as i.i.d. often yields better results than analyzing it as time series, frustratingly, as in this paper:

Multi-subject search correctly identifies causal connections and most causal directions in the DCM models of the Smith et al. simulation study. NeuroImage, 58(3), 838-848.

This paper also treats the distributions as a mixture, though doesn't assume non-i.i.d.

3.	As a result, it seems that any study proposing a time series analysis of this data should do a comparison of this result to one obtained by treating the data as i.i.d. instead, since this is a known phenomenon for this particular dataset. This is an issue because the proposed method is specifically designed to deal with time series.
4.	The problem with the DREAM3 examples is that none of the reported results have any lift with AUC; they’re all stuck around 0.5, which is more or less random. The differences between the methods are slight. This result doesn’t give a case where the proposed method is particularly helpful.

**Questions:**

Can you find better examples to show the usefulness of the theory?

Where can the theory really shine so far as the empirical application is concerned? Where can it be expected to do significantly better than alternative methods?

---

> ### Author Response · Authors · 2023-11-21
>
> We express our gratitude to the reviewer for their valuable feedback and inquiries. In the following section, we will tackle the primary issues raised.
>
> # Weaknesses
>
> > If we turn to the experimental section of the paper, we get two examples: one for NetSim (which I’m very familiar with) and another for the DREAM3 gene network. Both of these have problems with respect to the goal of this paper, suggesting that the choice of experimental datasets could be improved.
>
> We found it quite challenging to find large-scale real-world experimental datasets for time series causal discovery. If you  have any suggestions or pointers for datasets, we would be glad to run experiments on them and incorporate them into our draft.
>
> > The problem with the NetSim data is that it’s not an extremely convincing time series, as the records in the simulation are spaced far enough apart in time to render the data nearly i.i.d. In fact, analyzing it as i.i.d. often yields better results than analyzing it as time series, frustratingly, as in this paper:
>
> > Ramsey, J. D., Hanson, S. J., & Glymour, C. (2011). Multi-subject search correctly identifies causal connections and most causal directions in the DCM models of the Smith et al. simulation study. NeuroImage, 58(3), 838-848.
> > This paper also treats the distributions as a mixture, though doesn't assume non-i.i.d.
>
> We respectfully disagree. We note that several contemporary papers on time series causal discovery use this dataset to benchmark their methods [1, 2, 3].
> Furthermore, we examined the paper cited by the reviewer and discovered that the paper demonstrated that the IMaGES algorithm performed better than other contemporary i.i.d causal discovery algorithms; however, *they did not make a claim that analyzing it as i.i.d yields better results than analyzing it as time series*. Indeed, they did not compare against any temporal causal discovery baselines.
>
> > As a result, it seems that any study proposing a time series analysis of this data should do a comparison of this result to one obtained by treating the data as i.i.d. instead, since this is a known phenomenon for this particular dataset. This is an issue because the proposed method is specifically designed to deal with time series.
>
> Following your suggestion, we benchmarked well-known i.i.d causal discovery algorithms against simulation 3 data from the NetSim dataset. We were unable to run the IMaGES algorithm from [4] since we could not find a publicly available implementation. We compare the performance against reported numbers from temporal causal discovery methods in literature. The results are summarized below:
>
> | **Method** | **Orientation F1** | **Orientation AUROC** |
> |:----------:|:------------------:|:---------------------:|
> |     PC     |        0.433       |         0.689         |
> |   LiNGAM   |        0.050       |         0.413         |
> |     GES    |        0.414       |         0.679         |
> |  VARLiNGAM |        0.603       |         0.781         |
> |  DYNOTEARS |        0.615       |         0.946         |
> |   PCMCI+   |        0.583       |         0.771         |
>
> We observe that the temporal causal discovery algorithms handily outperform the i.i.d causal discovery baselines.
>
> > The problem with the DREAM3 examples is that none of the reported results have any lift with AUC; they’re all stuck around 0.5, which is more or less random. The differences between the methods are slight. This result doesn’t give a case where the proposed method is particularly helpful.
>
> DREAM3 is a challenging dataset due to the low number of samples (46 samples per graph) in the dataset. Nevertheless, we quite significantly outperform the baselines. More importantly, the clustering accuracy is high (94.83%) as reported in Section 5.3, which means that despite being unable to infer the causal relationships themselves, our method can distinguish among data coming from different causal models despite the limited samples. We believe that this is most definitely a significant outcome.

---

> > ### Comment · Reviewer_dKxB · 2023-11-21
> > **Still disagree...**
> >
> > I guess I just disagree about the DREAM dataset; the AUC of 0.555 is not high enough to predict much of anything well. I. still think this was not a good example to use.
> >
> > I believe the comparison of i.i.d. to non-i.i.d. was taken up by previous papers in NeuroImage.
> >
> > As for public availability, I looked at the paper I referenced after you said it, and now I think I may see the problem. I realized the link to the Tetrad project that contained that algorithm at the time, in the paper was a dead link. But I did a quick Google search for "tetrad github" and came up with this:
> >
> > https://github.com/cmu-phil/tetrad
> >
> > It looks like digging into the files, the IMaGES algorithm is still there.
> >
> > https://github.com/cmu-phil/tetrad/blob/development/tetrad-lib/src/main/java/edu/cmu/tetrad/algcomparison/algorithm/multi/Images.java
> >
> > The project has been very recently updated, and it looks like the implementation of IMaGES was changed earlier this year., so it may not be exactly the version that was used in that paper, perhaps a drawback, though it looks like it's along the same lines, just differently implemented. It still uses a GES algorithm and averages BIC scores from different samples.
> >
> > The Tetrad project looks to be a very public project (all code is public) that apparently dates back years, though I couldn't tell, looking at it quickly, how far back the repository goes. It is in Java, which can be a drawback, though there is a Python wrapper for the project that looks usable.
> >
> > I didn't mean to send you on a wild goose chase--just commenting that the IMaGES algorithm was reported in NeuroImage (a top journal for neuroscience methods at the time and still) for analysis of exactly the same data as you're using here, as handling mixtures of exactly the same sort as you are trying to handle, though for the i.i.d. case--following up on the seminal Smith et al. paper that introduced the NETSIM dataset in the first place-- i.e. this paper:
> >
> > Smith, Stephen M., et al. "Network modelling methods for FMRI." Neuroimage 54, no. 2 (2011): 875-891.
> >
> > So the idea of dealing with these particular mixtures is not novel--though the idea of doing it in a temporal setting is, I think. And then the question is, does the temporal setting buy you anything with this particular example? I do think it's relevant.

---

> ### Author Response · Authors · 2023-11-21
>
> # Questions:
> >Can you find better examples to show the usefulness of the theory?
>
> Identifiability is a cornerstone theory in causal discovery. Without it, any empirical case studies would be meaningless. That is, if there is no guarantee that there exists an underlying true set of graphs that our algorithm can find, then even strong empirical performance would be in vain. We would like to point out that the theory results presented in the paper pertain to the question of structural identifiability of the mixture model, which is a general statistical question rather than pertaining to our model specifically.
> Identifiability results specify conditions under which the constituent causal graphs can be uniquely determined from the mixture distribution.
>
> > Where can the theory really shine so far as the empirical application is concerned? Where can it be expected to do significantly better than alternative methods?
>
> Although not particularly pertaining to the theoretical results presented in the paper, we would expect our method to work better than alternative methods when the underlying causal graphs are sufficiently diverse and different from each other.
>
> References:
> [1] Gong, Wenbo, et al. "Rhino: Deep causal temporal relationship learning with history-dependent noise." arXiv preprint arXiv:2210.14706 (2022).
> [2] Cheng, Yuxiao, et al. "CUTS: Neural Causal Discovery from Irregular Time-Series Data." arXiv preprint arXiv:2302.07458 (2023).
> [3] Löwe, Sindy, et al. "Amortized causal discovery: Learning to infer causal graphs from time-series data." Conference on Causal Learning and Reasoning. PMLR, 2022.
> [4] Ramsey, J. D., Hanson, S. J., & Glymour, C. (2011). Multi-subject search correctly identifies causal connections and most causal directions in the DCM models of the Smith et al. simulation study. NeuroImage, 58(3), 838-848.

---

### Official Review · Reviewer_vYSG · 2023-11-01

**Soundness:** 2 fair
**Presentation:** 3 good
**Contribution:** 2 fair
**Rating:** 3
**Confidence:** 4

**Summary:**

This work presents a method that infers causal relationships from time-series data by allowing for mixtures of different causal models, rather than assuming a single underlying causal model. The authors utilize end-to-end variational inference to optimize parameters and perform inference on causal graphs, functional equations, and sample membership within mixture components. The method is assessed on both synthetic and real-world datasets, showcasing competitive performance on training data in causal discovery tasks. The authors establish the identifiability of the proposed model under mild assumptions.

**Strengths:**

The paper addresses a relevant and interesting aspect of causal discovery in time-series data.

The proposed loss function is a simple extension of the standard variational inference objective, enabling efficient end-to-end training with a mixture of core causal discovery models.

The method is flexible in terms of the choice of core causal structure learning algorithms, inheriting the structural identifiability properties of these algorithms.

Competitive empirical results (although on training data) are reported across various metrics.

**Weaknesses:**

The proposed objective function can be seen as a straightforward extension of the standard variational inference optimisation framework. Therefore, the overall novelty and significance of the work may be somewhat limited in this regard.

A major drawback of the work is that the reported results are based on training data, as the proposed method depends on learnable sample-specific parameters. It raises questions about why an encoder producing a K-way categorical random variable given a sample was not considered by the authors.

The lack of reported results on generalisation limits insight into the proposed method's ability to perform beyond the training data.

Although the method claims flexibility in terms of core causal structure learning algorithms, performance is only demonstrated based on one causal discovery method.

**Questions:**

Can you please clarify how the AUROC metric is computed for the different methods being compared?

Regarding the experiment on Netsim-permuted data, wouldn’t one expect the results of the proposed method to be inherently favorable compared to non-mixture alternatives? This is because by permuting the variables, we intentionally manipulate the data to explicitly align with the underlying assumption of a mixture model.

What specific hyper-parameter value K was chosen for the DREAM3 Gene Network experiment?

---

> ### Author Response · Authors · 2023-11-21
>
> We appreciate the reviewer for their valuable feedback and inquiries. In the subsequent section, we will address the key concerns that have been raised.
>
> # Weaknesses:
> > The proposed objective function can be seen as a straightforward extension of the standard variational inference optimisation framework. Therefore, the overall novelty and significance of the work may be somewhat limited in this regard.
>
> We politely disagree. The problem of inferring multiple causal graphs is quite challenging and has received scant attention in the literature despite its widespread applicability. Our method makes significant contributions with (1) designing an effective deep latent variable model and deriving an ELBO objective to infer the relevant parameters (2) proving the identifiability conditions and (3) solid empirical results.  Even though the objective function is an instance of variational inference, it is not the novelty that we claim in this paper.  Despite its relative “simplicity”, it achieves competitive empirical results.
>
> > A major drawback of the work is that the reported results are based on training data, as the proposed method depends on learnable sample-specific parameters. It raises questions about why an encoder producing a K-way categorical random variable given a sample was not considered by the authors.
>
> There seems to be a misunderstanding of the problem. Causal discovery is an “unsupervised” problem, in the sense that the ground truth causal graph(s) are assumed to be unknown. Reporting causal discovery results on training data is a standard practice in the literature. We point out that many similar works on optimization-based causal discovery methods work on similar principles, e.g. [1,2,3] Our proposed method reframes the task to an optimization problem, which we solve by training neural networks. In other words, our inference procedure itself involves training on the input data $X$. We have clarified this in the Problem Setting in Section 3 of the paper.
>
> We learn sample-specific parameters, as the task involves inferring membership information solely for the samples within the input data $X$. There is no need for generalization to unseen points. Utilizing an "encoder producing a K-way categorical random variable" would essentially achieve the same objective of learning sample-specific parameters. Using a K-way encoder is a good suggestion that would work especially well for a large sample size. However, since all our experiments are run in the low-sample regime, sample-specific weights work sufficiently well with low memory footprint.
>
> > The lack of reported results on generalisation limits insight into the proposed method's ability to perform beyond the training data.
>
> As noted earlier, the causal discovery problem only involves inferring causal graphs for the training data $X$, and hence the notion of “generalization beyond the training data” is not applicable in this context.
>
>
> > Although the method claims flexibility in terms of core causal structure learning algorithms, performance is only demonstrated based on one causal discovery method.
>
> We focus on one “core” causal structure learning algorithm (i.e. Rhino [1]) in order to thoroughly evaluate its capabilities and performance. While other algorithms could potentially be more suitable for specific applications, we find that MCD with Rhino seems to work reasonably well for the tasks we selected and to demonstrate the utility of our framework.
>
> # Questions:
> > Can you please clarify how the AUROC metric is computed for the different methods being compared?
>
> For Rhino and MCD, which parameterize the causal graphs as Bernoulli distributions over each edge, we use the inferred edge probability matrix as the “score”, and evaluate the AUROC metric between the score matrix and the true adjacency matrix. For DYNOTEARS, we use the absolute value of the output scores and evaluate the AUROC. For PCMCI+ and VARLiNGAM, since they only output adjacency matrices, we directly evaluate the AUROC between the predicted and true adjacency matrices. We have updated Appendix D.1 with this clarification.
>
> > Regarding the experiment on Netsim-permuted data, wouldn’t one expect the results of the proposed method to be inherently favorable compared to non-mixture alternatives? This is because by permuting the variables, we intentionally manipulate the data to explicitly align with the underlying assumption of a mixture model.
>
> We performed random permutation of the nodes to demonstrate that even though methods that predict a single graph can perform reasonably well when the different causal models are similar, they are susceptible to poor performance under very mild “perturbations” i.e. permutation of the nodes.  Through node permutation, we generate a mixture distribution where the constituent causal graphs differ significantly. This process validates our hypothesis that the proposed method (MCD) should be effective in such scenarios.

---

> > ### Author Response · Authors · 2023-11-21
> >
> > > What specific hyper-parameter value K was chosen for the DREAM3 Gene Network experiment?
> >
> > We used $K=10$. We have updated the draft to specify this choice.
> >
> > References:
> > [1] Gong, Wenbo, et al. "Rhino: Deep causal temporal relationship learning with history-dependent noise." arXiv preprint arXiv:2210.14706 (2022).
> > [2] Zheng, Xun, et al. "Dags with no tears: Continuous optimization for structure learning." Advances in neural information processing systems 31 (2018).
> > [3] Zantedeschi, Valentina, et al. "DAG Learning on the Permutahedron." arXiv preprint arXiv:2301.11898 (2023).

---

### Official Review · Reviewer_vKpF · 2023-11-19

**Soundness:** 3 good
**Presentation:** 3 good
**Contribution:** 2 fair
**Rating:** 5
**Confidence:** 3

**Summary:**

Authors propose a variational inference adopted for causal discovery in time-series data; in particular for mixtures of multiple causal graphs.

**Strengths:**

- in general, exposition is good (although there is room for clarity and explanations in formal parts ).
- addressing  mixture of multiple-causal graph  discovery is an interesting/relevant direction of research, that hasn't been much investigated (although I have my reservations)
- experimental results are coupled with a theoretical structural identifiability result.

**Weaknesses:**

- my main skepticism is due to the fact that all the results are for the training set, which I am surprised (and had a stronger positive impression until that point). Trivial enough: For an unquestionable positive score, I would rather need result on test data, on unseen graphs.

-   How different the causal graphs in these mixture models (and also the data i.e., average SHD within the cluster is missing.  This is really important to really understand what is going on behaviour of the method. (hence, my question). No surprise to see its effect on highly-imbalanced one.

- on minor point,  an illustrative toy example is lacking would be very useful.

**Questions:**

-  What is the effect of the distance between causal graphs to the performance?

-  g_1 and g_2 are not defined in Theorem 1, is it a typo: should be h? or a particular graph?

---

> ### Author Response · Authors · 2023-11-21
>
> We would like to thank the reviewer for their valuable feedback and thoughtful inquiries. Below, we respond to the main concerns that have been raised.
>
> # Weaknesses:
> > my main skepticism is due to the fact that all the results are for the training set, which I am surprised (and had a stronger positive impression until that point). Trivial enough: For an unquestionable positive score, I would rather need result on test data, on unseen graphs.
>
> As pointed out in the reply to Reviewer vYSG, we reiterate that this is standard practice in causal discovery, where the causal graph is inferred without any supervision. We do not assume access to a training corpus of ground-truth graphs or functional relationships; as such, all results are on “unseen graphs”. We have clarified this in the Problem Setting in Section 3 of the paper.
>
> > How different the causal graphs in these mixture models (and also the data i.e., average SHD within the cluster is missing. This is really important to really understand what is going on behaviour of the method. (hence, my question). No surprise to see its effect on highly-imbalanced one.
>
> Within the cluster, the average SHD would be 0, since we predict the same graph for each cluster. We plot the average pair-wise SHD (over all pairs of causal graphs) for each dataset here:
> | Dataset         | D   | K* | Mean SHD | Std Dev SHD | Min SHD | Max SHD |
> |-----------------|-----|----|----------|-------------|---------|---------|
> | Synthetic       |   5 |  5 |    24.00 |        1.95 |      20 |      27 |
> | Synthetic       |   5 | 10 |    23.24 |        3.37 |      14 |      30 |
> | Synthetic       |   5 | 20 |    22.61 |        3.29 |      13 |      31 |
> | Synthetic       |  10 |  5 |    53.40 |        3.83 |      48 |      59 |
> | Synthetic       |  10 | 10 |    54.09 |        3.26 |      45 |      61 |
> | Synthetic       |  10 | 20 |    54.27 |        3.73 |      44 |      64 |
> | Synthetic       |  20 |  5 |   113.80 |        5.62 |     101 |     120 |
> | Synthetic       |  20 | 10 |   111.91 |        5.42 |      99 |     123 |
> | Synthetic       |  20 | 20 |   113.59 |        5.06 |      98 |     124 |
> | DREAM3          | 100 |  5 |   517.60 |      202.13 |     234 |     896 |
> | Netsim          |   5 | 14 |     2.59 |        1.17 |       1 |       5 |
> | Netsim-permuted |  15 |  3 |    34.00 |        1.63 |      32 |      36 |
> We have also added this table to the paper in Appendix D.4
>
> > on minor point, an illustrative toy example is lacking would be very useful.
>
> Thank you for the suggestion. We have added an illustrative toy example with two causal graphs in Appendix E to explain the importance of using a mixture model for heterogeneous causal discovery.
>
> # Questions
> > What is the effect of the distance between causal graphs to the performance?
>
> We believe that our model is better able to cluster points when the SHD between causal graphs is larger. This is especially prominently seen with the Netsim and Netsim-permuted datasets.
>
> > g_1 and g_2 are not defined in Theorem 1, is it a typo: should be h? or a particular graph?
>
> Yes, it is a typo. They should be $h_1$ and $h_2$. This has been fixed in the new draft.

---

### Author Response · Authors · 2023-11-21

We thank all the reviewers for the detailed comments, suggestions, and questions about our paper. We are glad that they recognized the broad applicability of our problem setting (vYSG, dKxB, bLPE, vKpF) and appreciated our extensive experimental results and ablation studies (vYSG, bLPE, vKpF), and theoretical results (bLPE, vKpF). Based on the reviewer’s suggestions, we have updated the draft to include several additional experiments:

1. We have re-run all experiments with the synthetic datasets for 5 runs.
2. We have added Appendix B.2 to show the clustering accuracy for different values of $D$.
3. We have added Appendix B.3 to show the progression of clustering with training.
4. We have added Rhino (grouped) results for the synthetic datasets (Figure 3) and ablations with different number of underlying graphs (Figure 6) and using true graph indices (Figure 7).
5. We have added an illustrative toy example in Appendix E.

We have highlighted all the changes made in the draft in blue color to make them easier to spot.

---

### Meta-Review · Area_Chair_5q8u · 2023-12-06

**Metareview:**

The paper proposes to learn a mixture of SCMs from time data, via an EM style approach.

Strengths: This is an interesting problem and theoretical insights are presented.

Weaknesses: The reviewers had several points about methodological novelty, exposition of the paper, experimental evaluation, etc.

**Justification For Why Not Higher Score:**

The reviewers had several points about methodological novelty, exposition of the paper, experimental evaluation, etc, which make the paper in its current form inapt for ICLR.

**Justification For Why Not Lower Score:**

NA

---

### Decision · Program_Chairs · 2024-01-16

Reject